# EEG-EyeTrack: A Benchmark for Time Series and Functional Data Analysis with Open Challenges and Baselines

## Abstract

We present a new benchmark dataset for functional data analysis (FDA), focusing on the reconstruction of eye movements from EEG data. Our contribution is threefold: first, we introduce a challenging dataset collected with consumer-grade hardware under realistic conditions. Second, we propose open challenges and evaluation metrics tailored to FDA applications. Third, we establish baseline results for the primary regression task of reconstructing eye movements from EEG signals using functional neural networks. We report baseline results on both our new dataset and the established EEGEyeNet dataset, which was recorded with research-grade hardware.

## 1 Introduction

In the future, brain-computer interfaces (BCIs) might offer the possibility of restoring or augmenting sensory perception, such as allowing a blind person to see (Muqit et al., 2024), or even converting thoughts into text (Willett et al., 2023). Currently, electroencephalography (EEG), the technology behind modern BCIs, is used primarily to assist in the diagnosis of neurological diseases (Behzad & Behzad, 2021; Britton et al., 2016). Automated analysis of EEG data is an active area of research. However, there is still a long way to go before brain activity can be reliably analyzed, and used for the development of robust BCIs. A more feasible application of BCIs is EEG-based eye tracking Dietrich et al. (2017); Kastrati et al. (2021); Fuhl et al. (2023); Sun et al. (2023). Typically, electric activity related to eye movements is ignored or filtered out (Croft & Barry, 2000), but it might be used to reconstruct the gaze direction. A reliable and accurate tracking of eye movement opens up new possibilities for BCIs. EEG-based eye tracking has the advantage of requiring no additional hardware when brain activity is already being monitored with an EEG headset. Additionally, it remains effective even when the eyes are closed, such as during sleep. However, current EEG-based eye trackers are mainly based on EEG data recorded with expensive hardware in a laboratory setting. Open datasets recorded with consumer-level hardware outside a lab environment are scarce, yet crucial for the development of EEG-based eye trackers that work reliably in precisely this context.

Two promising fields of statistics for the interpretation of EEG data are time series and functional data analysis (FDA). FDA provides statistical tools for data that may be modelled as smooth functions (of time, space, frequency or any other reasonable argument), see Ramsay & Silverman (2005) and Kokoszka & Reimherr (2017) for foundational overviews. Recent extensions into deep learning give rise to functional neural networks (FNNs), which embed such smoothness directly into network layers (see Rossi et al., 2002; Rao & Reimherr, 2023b; Heinrichs et al., 2023, among others). It can be assumed that developments in these areas will enable robust and accurate eye tracking. However, not all FDA benchmark datasets are equally suited for modern methods. Many are either too simple, such as the Tecator dataset (Thodberg, 2015) with perfect classification accuracy, or unsolvable due to class overlap, such as the Phoneme dataset (Hastie et al., 1995). The aim of this work is two-fold. On the one hand, we introduce the *Consumer-Grade EEG-Based Eye Tracking Dataset* as a new, challenging benchmark for methods in the areas of time series analysis and FDA. For this purpose, open problems and evaluation metrics are established. On the other hand, we investigate the use of methods from FDA for the central problem of eye movement reconstruction from EEG data, which can be considered a scalar-on-function or function-on-function regression problem, and give first baseline results.

To obtain the baseline results, an existing model for eye movement reconstruction, as proposed by Fuhl et al. (2023), was used as a reference. Recently, neural networks with functional layers were specifically designed for the analysis of EEG data. These functional neural networks (FNNs) are expected to better model the smoothness of the data, and therefore require fewer parameters and be more robust to noise compared to other models. As FNNs have not been used for EEG-based eye tracking, we compare the models also on the already existing EEGEyeNet dataset.

The majority of FDA methods requires previous alignment of curves, which is referred to as *function registration*. However, this is only possible if meaningful features can be identified in the data or if events of interest are externally triggered. In the case of EEG data, where the event of interest is often a self-paced action and the signal is noisy and complex, curve registration is infeasible. Therefore, EEG-based eye tracking data may be of interest for developing and testing new methods that work with unregistered data. However, due to the experimental design, the true eye movement data is known and can serve as a basis for curve alignment. This allows methods that require function registration to be applied to the dataset as well.

To summarize, our contribution is as follows:

- **Challenging dataset:** We introduce and publicly release the "Consumer-Grade EEG-Based Eye Tracking Dataset", a large-scale collection of EEG and eye-tracking recordings acquired with consumer-grade hardware under realistic, non-laboratory conditions.

- **Standardized FDA benchmark:** We formalize scalar-on-function regression on the public "Consumer-Grade EEG-Based Eye Tracking Dataset" with standardized evaluation metrics for reproducible FDA benchmarking.

- **Baseline performance:** We provide initial benchmarks by evaluating different functional and conventional models, on both the consumer-grade dataset and the research-grade EEGEyeNet, establishing reference numbers.

## 2 Related Work

Typically, when recording EEG data, one is interested in brain activity rather than artifacts from muscle or eye movements. The latter are either filtered out, with methods such as independent component analysis (ICA) or canonical correlation analysis (CCA), or completely ignored, especially in deep learning-based pipelines (Urigüen & Garcia-Zapirain, 2015). Recently, however, the reconstruction of eye movements from EEG recordings has been studied in the literature as an independent task. Dietrich et al. (2017) recorded short segments of EEG data, containing extreme eye movements (left, right, up, down), with an EEG headset with 14 wet electrodes. They classified the recordings with a variant of $k$-nearest neighbors, reaching an accuracy of 96%. Subsequently, Sun et al. (2023) recorded eye movements in a laboratory setting with 64 wet electrodes. The proposed algorithm, EEG-VET, was able to reconstruct *saccadic* (rapid) and smooth eye movements. Recently, the EEGEyeNet dataset was introduced as a benchmark dataset for the problem of reconstructing eye movement from EEG recordings (Kastrati et al., 2021). The dataset contains recordings from 356 subjects, comprising 38 hours of saccadic left-right eye movements, 7 hours and 52 minutes of saccadic movements on a grid, and 1 hour and 29 minutes of free eye movements. The EEG data was recorded in a laboratory setting with an EEG headset containing 128 electrodes. Based on the EEGEyeNet, Fuhl et al. (2023) proposed a neural network architecture, where the first layer should act as a (learnable) spatial filter. This model, referred to as *SpatialFilterCNN* in the remainder of this work, led to new state-of-the-art results for the EEGEyeNet dataset and is used as benchmark for subsequent experiments. Our newly introduced dataset is similar to EEGEyeNet, yet recorded with consumer-grade hardware outside a controlled laboratory setting, that shall allow the development of EEG-based eye trackers in real-world applications. The used EEG-headset had only four dry electrodes, and therefore a substantially lower signal-to-noise ratio. We will refer to our dataset as "Consumer-Grade EEG-based Eye Tracking" dataset.

Functional data analysis (FDA), a field of statistics, deals with smooth processes. If eye movements are recorded with a sufficiently high sampling frequency, the resulting data can be regarded as a smooth function of time. A variety of open datasets for different tasks are commonly used as benchmarks. These datasets

include the "Berkeley Growth Study" (Tuddenham, 1954), daily temperatures from Canadian weather stations and the "Handwriting" dataset (Ramsay & Silverman, 2005). For classification of functional data, the "Phoneme" and "Tecator" datasets are frequently used (Hastie et al., 1995; Thodberg, 2015). Especially for the latter two, very high accuracies ($> 91\%$ and $100\%$, respectively) have been achieved in the literature (Heinrichs et al., 2023). For future developments in FDA, new and challenging datasets are required. Based on our "Consumer-Grade EEG-based Eye Tracking" dataset, we formulate multiple open challenges in FDA in Section 4.1.

Most FDA methods require data to be *registered*, which means aligning the (functional) data to a common time axis, where each time point contains essentially the same information across all functions (Kneip & Gasser, 1992; Gasser & Kneip, 1995; Ramsay & Li, 1998). This does not only include classic methods, such as functional PCA (FPCA) and functional linear models (Shang, 2014; Cardot et al., 1999; Cuevas et al., 2002), but also modern methods for functional time series, such as tests on stationarity or white noise (Bücher et al., 2020; 2023).

Although curve registration is a common preprocessing step, it is not feasible in many applications, especially when sliding windows are considered. Therefore, different shift-invariant methods have been proposed recently, such as transform-invariant FPCA (Heinrichs, 2025). For classification and regression tasks, functional neural networks have been proposed and extensively studied (Rossi et al., 2002; Rossi & Conan-Guez, 2005; Rossi et al., 2005). Early functional neural networks essentially project the (infinite-dimensional) functions onto multivariate vectors in the first layer and use "standard" layers throughout the remainder. More recently, fully functional neural networks have been proposed for scalar-on-function and function-on-function regression (Rao & Reimherr, 2023a;b). Additionally, convolutional layers have been extended to functional data (Heinrichs et al., 2023). In the remainder, we will use the definitions of functional neurons from the latter reference, as it has been applied to EEG data in the past, and refer to it for explanations of the functional layers and their hyperparameters.

## 3 Functional Data Analysis

Functional data analysis (FDA) offers a framework for modeling signals, such as EEG recordings, as smooth functions $x(t)$ rather than high-dimensional vectors of discrete time points. With this approach, dependencies between neighboring points are explicitly taken into account. A first step in FDA, usually consists of smoothing the discrete observations to obtain smooth data. Afterwards, the functional data is often projected from an infinite-dimensional function space to a lower dimensional space by dimension reduction techniques, such as functional principal component analysis (FPCA), so that multivariate methods can be used for a subsequent analysis.

Through dimension reduction, important information might get lost, and it can be advantageous to work directly with functional data. Depending on the analysis' goal, it is often assumed that the observations are elements of $C([0, 1])$, the Banach space of continuous functions on $[0, 1]$, or elements of $L^2([0, 1])$, the Hilbert space of square-integrable functions on $[0, 1]$.

Let $H = L^2([0, 1])$ denote the function space of the data. A central problem of FDA, is the task of regression, that links a functional input to a scalar or functional response. In *scalar-on-function* regression, a scalar outcome $y$ is modeled as $y = F(x) + \epsilon$, where $F : H \to \mathbb{R}$ denotes a (not necessarily linear) functional from the function space $H$ to $\mathbb{R}$. When $F$ is assumed to be continuous and linear, we obtain the linear model

$$y = \alpha + \int \beta(t)x(t)\mathrm{d}t + \epsilon,$$

where $\beta(t)$ is a coefficient function. *Function-on-function* regression generalizes this to predict an entire output curve $y(t)$, e. g., a gaze trajectory. In this case, $y = F(x) + \epsilon$, where $F : H \to H$ denotes an operator. Again, if $F$ is continuous and linear, we obtain the linear model

$$y(t) = \alpha(t) + \int \beta(t, s)x(s)\mathrm{d}s + \epsilon(t).$$

Functional neural networks (FNNs) allow modelling of non-linear relations between input and output. In these networks, discrete sums and convolutions are essentially replaced by their continuous counterparts. For input functions $x_1, \ldots, x_p$, defined on $[0, 1]$, a functional neuron might be defined by

$$y(t) = \sigma\left(w_0(t) + \sum_{i=1}^{p} w_i(t)x_i(t)\right),$$

for functional weights $w_0, \ldots, w_p$. Further, we may extend $x_1, \ldots, x_p$ to $[-b, 1+b]$, for some bandwidth $b > 0$, by defining the functions as zero in $[-b, 0) \cup (1, 1+b]$. Then, we can define a functional convolutional layer by

$$y(t) = \sigma\left(w_0(t) + \sum_{i=1}^{p} \int_{-b}^{b} w_i(s)x_i(t-s)\mathrm{d}s\right),$$

for functional weights $w_0 : [0, 1] \to \mathbb{R}$ and $w_1, \ldots, w_p : [-b, b] \to \mathbb{R}$, see Heinrichs et al. (2023). In the following, we consider smooth (continuous or differentiable) weights $w$. Due to the integration, the output becomes increasingly smooth. Generally, for differentiable weight functions $w_1, \ldots, w_p \in C^1([0, 1])$ and $k$-times differentiable inputs $x_1, \ldots, x_p \in C^k([0, 1])$, $y$ will be $k + 1$ times differentiable. Moreover, for bounded functions $x_1, \ldots, x_p$, $y$ is continuous and for continuous functions $x_1, \ldots, x_p$, $y$ is differentiable. This observation is crucial for the use of functional neural networks, and explains how deep FNNs filter out local noise.

Beyond regression, FDA encompasses tasks such as classification, clustering curves by shape, and detecting change points in functional time series. In this paper, we focus empirically on scalar-on-function regression, while defining open challenges related to other FDA tasks.

## 4 Dataset and Open Challenges

We introduce the "Consumer-Grade EEG-based Eye Tracking" dataset, which is publicly available on Zenodo[1]. The dataset contains recordings from 116 sessions of 113 participants. Each session lasted approximately 6 minutes, yielding a total of 11 hours and 45 minutes. The dataset is split into training (90%) and test (10%) sets subject-wise, ensuring no overlap between subjects in train and test sets.

The experiments consisted of a target moving on screen that participants were asked to follow closely. Data from three different modalities was recorded. First, EEG-data was measured at positions TP9, TP10, AF7, AF8 according to the international 10-10 system, where the electrode Fpz was used as reference. Second, the current gaze position, as estimated from a webcam-based eye tracker, was recorded. And finally, the target's position on screen was tracked.

Each session consisted of four stages. In the first two stages ("level-1" recordings), the target moved only horizontally and vertically on screen, while in the latter two stages ("level-2" recordings), the target moved in more directions, yielding more degrees of freedom. Each level had one stage, where the target changed its position abruptly, and another stage, where the target moved continuously across the screen. The former recordings are referred to as "saccades" while the latter as "smooth". Especially, recordings with smooth eye movements are of interest as benchmark for methods in functional data analysis.

Recorded with consumer-grade hardware (4 dry electrodes) in non-laboratory conditions, the dataset reflects realistic but challenging conditions for EEG-based eye tracking. In total, 14 out of 116 recorded sessions (12.1%) were flagged as having known quality issues (see Section B.3.1). These recordings from participants 2, 4, 16-20, 62-67, 79 and the "level-1-saccades" recording from participant 50 were excluded from downward analysis. Across all 4 EEG channels and the non-excluded 407 recordings, missing samples due to hardware dropouts ranged from $0.0\,\%$ to $24.1\,\%$ per channel (median $6.3\,\%$), with 27 recordings exceeding 10% of missing data (see Table 1). To preserve both non-stationary trends and the strong $50\,\mathrm{Hz}$ background noise, we imputed missing values with a SARIMA model with seasonal lag 5 and the remaining parameters tuned based on the Akaike Information Criterion. For an explanation of this process, as well as details regarding

---

[1]Link to non-anonymous repository

the experiments and data acquisition, we refer to Appendix B. For the benchmark experiments, we used the preprocessing as described therein. More specifically, notch filters at 50 Hz and 60 Hz were used, followed by a bandpass filter between 0.5 Hz and 40 Hz. The transition from the stopband to the passband of the bandpass filter allows some high-frequency noise to persist. This is mitigated by applying the notch filters beforehand.

| Statistic | Value (%) |
|---|---|
| Minimum missing per channel | 0.0 |
| Median missing per channel | 6.3 |
| Maximum missing per channel | 24.1 |
| Recordings with >10% missing | 27 |

Table 1: Summary of missing-data rates across EEG channels and recordings.

For details regarding participant demographics, recording environment, ethics/consent procedures, licensing conditions, and additional information, we refer to Appendix B.

In accordance with the guidelines of the German Research Foundation (DFG), no ethics vote was required for the existing study (German Research Foundation, 2023). More specifically, the DFG states that "A statement by an ethics committee is not usually required for studies involving electrophysiological leads (e.g. EEG, MEG, NIRS) if the above points do not apply.", where the "above points" refer to experiments involving high levels of stress, patients, (f)MRI or psychopharmacological studies; none of which applies here. In accordance with the GDPR, the EEG recordings were anonymized so that no conclusions can be drawn about the study participants.

However, the dataset used contains EEG signals that could potentially be misused for applications such as covert attention tracking or surveillance. Although the dataset itself does not include any personal identifiers, we caution against such misuse.

### 4.1 Open Challenges

Due to the experimental design, the dataset is well suited to serve as a benchmark for FDA methods. The main challenge of this dataset is the prediction of the target's position from the EEG signal. This can be formulated as a **function-on-function** or as a **scalar-on-function regression** problem, where in the first case the position over time and in the latter case the final position of the target should be predicted from the EEG data for (sliding) windows of a given length. Note that instead of the target's position, the gaze position, as estimated from a webcam-based eye tracker, might be used as a target variable as well.

The dataset can be used as a benchmark for FDA methods that require function registration, as well as for methods that do not require it. In level-1 experiments, the target starts in the center of the screen and moves up, down, left or right. The start of each movement can be used as a marker to divide an entire recording into multiple trials. These trials can be assumed to be aligned curves (or further registered). For methods that do not require registration, sliding windows can be generated from the entire recording.

For benchmark experiments, the originally proposed train-test split (90% training, 10% test) should be kept, and test data from other tasks/levels should not be used for training. For regression, both coordinates of the target, as defined in the columns `Stimulus_x` and `Stimulus_y` should be predicted. As a metric to measure the quality of predictions, we propose the Mean Euclidean Distance (MED) between the predicted and the ground truth stimulus position. The MED is defined as

$$\text{MED}_k = \frac{1}{N_k} \sum_{i=1}^{N_k} \sqrt{(x_i - \hat{x}_i)^2 + (y_i - \hat{y}_i)^2},$$

where $N_k$ denotes the number of samples in the $k$-th recording of the test data, $x_i$ and $y_i$ are the true, and $\hat{x}_i$ and $\hat{y}_i$ are the predicted target positions at time step $i$. The final score of a model is the MED over all

12 recordings in the test data, which is defined as

$$\text{MED} = \frac{1}{\sum_{k=1}^{12} N_k} \sum_{k=1}^{12} N_k \text{MED}_k.$$

The MED is a standard metric in eye tracking, because of its easy and clear interpretation (Raghunath et al., 2012; Papoutsaki et al., 2018; Dalrymple et al., 2018). It measures the average distance between the ground truth stimulus position and its corresponding prediction. Other possible metrics are the MSE, RMSE, or MAE, but compared to the MED they do not result in values that are easily interpretable, as they correspond to the mean of their respective one-dimensional metrics for the $x$- and $y$-axis. For easier comparison with other datasets, we also report the RMSE, defined analogously to the MED via the recording-level RMSE

$$\text{RMSE}_k = \left( \frac{1}{N_k} \sum_{i=1}^{N_k} (x_i - \hat{x}_i)^2 + (y_i - \hat{y}_i)^2 \right)^{1/2}$$

as

$$\text{RMSE} = \left( \frac{1}{\sum_{k=1}^{12} N_k} \sum_{k=1}^{12} N_k \text{RMSE}_k^2 \right)^{1/2}$$

When reporting the performance of a model, the MED for each task should be reported. This allows for a more nuanced comparison of the performance of different methods, as models could overfit to one task and perform poorly on others, making them unsuitable for general purpose eye tracking.

A schematic overview of the benchmarking pipeline is given in Figure 1 (a).

Besides the main challenge of (scalar-on-function) regression, we identified several additional challenges:

1. **Classification of Movements:** Classify EEG data based on eye movements, e. g., "horizontal" vs. "vertical", "saccades" vs. "smooth", or "up"/"down"/"left"/"right".

2. **Classification of Participants:** Classify participants into different groups with class labels generated from eye movements, e. g., as good or poor trackers, based on the difference between target and gaze position; or as fast or slow trackers, based on the lag between target and gaze position.

3. **Clustering:** Identify brain activity patterns from EEG signals.

4. **Dimension Reduction:** Reduce the dimension of the data while minimizing the reconstruction loss.

5. **Outlier detection:** Identify segments, or time points, with unusual data, e. g. due to missing values that have been replaced by zeros or other erroneous measurements.

6. **Change point detection:** Detect moments where gaze tracking or EEG activity shifts significantly, e. g., due to a loss of attention or transitions between movements and pauses.

## 4.2 Dependence Analysis

Due to the previously mentioned hardware limitations, it is unclear whether the recorded data contains sufficient information to solve the open challenges. Modest baseline results might be due to the selected model or inherent due to the data. In order to measure the dependence between gaze/stimulus and EEG, we calculated Chatterjee's correlation coefficient (Chatterjee, 2021). For some regressor $X \in \mathbb{R}^d$ and a response variable $Y \in \mathbb{R}$, Chatterjee's $\xi$ has the property that $\xi(Y, X) = 0$ if and only if $X$ and $Y$ are independent, and $\xi(Y, X) = 1$ if any only if $Y$ is (a.s.) $\sigma(X)$-measurable. To determine whether EEG and gaze/stimulus are (non-linearly) dependent, we set $X$ to be an EEG window of length $w \in \{1, 2, 4, 8, 16, 32, 64, 128, 256, 512\}$ and $Y$ to the $x$ or $y$ coordinate of the gaze/stimulus at the last time instant of the window. This measures

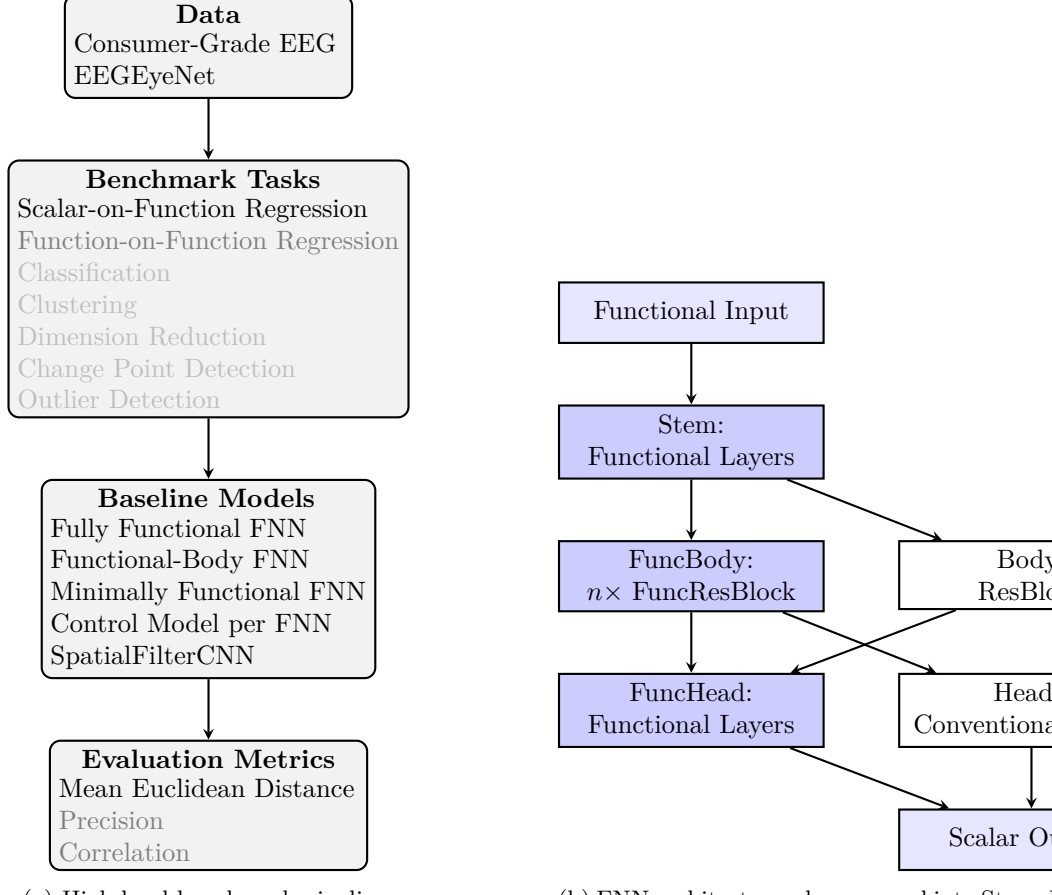

(a) High-level benchmark pipeline.

(b) FNN architectures decomposed into Stem, Body, and Head. Functional layers shaded blue; input/output layers shaded in light blue; conventional layers left white.

Figure 1: (a) Benchmark pipeline from datasets through tasks to evaluation. (b) Functional neural network variants: every FNN has a stem (functional), a body (functional or conventional), and a head (functional or conventional).

the required dependence for scalar-on-function regression. Table 2 shows the average values of $\xi$ across all recordings (Table 21 contains additionally standard deviations).

As expected, the (possibly non-linear) dependence between EEG and stimulus/target is low for short windows, but quickly increases as the context window grows. For windows of length 256 and 512 (corresponding to 1s and 2s), the (non-linear) correlation is already close to 1. Interestingly, the coefficient has similar values for the $x$- and $y$-coordinate. Moreover, the standard deviation (for large values of $w$) is approx. 0.3, indicating a high variation between recordings. Overall, Chatterjee's $\xi$ suggests that the EEG contains relevant information for the prediction of the gaze and stimulus. This indicates that the modest results are due to the selected models.

## 5 Model Architectures

Because FNNs are rather new and effective design practices are not yet established, we relied on established CNN architectures as a reference point to guide the development of our FNN architectures. These architectures typically consist of three main components: the *stem*, *body*, and *head*.

| $Y \quad \backslash \quad w$ | 1 | 2 | 4 | 8 | 16 | 32 | 64 | 128 | 256 | 512 |
|---|---|---|---|---|---|---|---|---|---|---|
| *Panel A: level-1-saccades* | | | | | | | | | | |
| $x$-Gaze | 0.050 | 0.072 | 0.120 | 0.211 | 0.398 | 0.565 | 0.667 | 0.711 | 0.858 | 0.892 |
| $y$-Gaze | 0.049 | 0.051 | 0.074 | 0.141 | 0.317 | 0.501 | 0.619 | 0.749 | 0.862 | 0.868 |
| $x$-Stimulus | 0.060 | 0.077 | 0.121 | 0.208 | 0.393 | 0.564 | 0.668 | 0.713 | 0.875 | 0.888 |
| $y$-Stimulus | 0.051 | 0.054 | 0.079 | 0.147 | 0.322 | 0.504 | 0.620 | 0.749 | 0.888 | 0.868 |
| *Panel B: level-1-smooth* | | | | | | | | | | |
| $x$-Gaze | 0.067 | 0.094 | 0.151 | 0.252 | 0.443 | 0.611 | 0.702 | 0.779 | 0.790 | 0.925 |
| $y$-Gaze | 0.069 | 0.075 | 0.110 | 0.189 | 0.373 | 0.552 | 0.656 | 0.743 | 0.801 | 0.913 |
| $x$-Stimulus | 0.067 | 0.091 | 0.144 | 0.241 | 0.431 | 0.601 | 0.691 | 0.770 | 0.776 | 0.917 |
| $y$-Stimulus | 0.064 | 0.070 | 0.103 | 0.179 | 0.366 | 0.547 | 0.651 | 0.741 | 0.784 | 0.919 |
| *Panel C: level-2-saccades* | | | | | | | | | | |
| $x$-Gaze | 0.074 | 0.109 | 0.170 | 0.273 | 0.441 | 0.588 | 0.692 | 0.757 | 0.837 | 0.855 |
| $y$-Gaze | 0.080 | 0.088 | 0.124 | 0.196 | 0.364 | 0.525 | 0.637 | 0.707 | 0.851 | 0.852 |
| $x$-Stimulus | 0.090 | 0.101 | 0.144 | 0.225 | 0.378 | 0.524 | 0.630 | 0.721 | 0.839 | 0.863 |
| $y$-Stimulus | 0.088 | 0.098 | 0.136 | 0.212 | 0.367 | 0.516 | 0.619 | 0.703 | 0.861 | 0.861 |
| *Panel D: level-2-smooth* | | | | | | | | | | |
| $x$-Gaze | 0.079 | 0.109 | 0.179 | 0.214 | 0.487 | 0.653 | 0.727 | 0.801 | 0.794 | 0.938 |
| $y$-Gaze | 0.099 | 0.104 | 0.153 | 0.229 | 0.426 | 0.594 | 0.690 | 0.779 | 0.822 | 0.920 |
| $x$-Stimulus | 0.076 | 0.089 | 0.134 | 0.240 | 0.417 | 0.579 | 0.678 | 0.766 | 0.797 | 0.919 |
| $y$-Stimulus | 0.082 | 0.087 | 0.126 | 0.290 | 0.397 | 0.559 | 0.657 | 0.751 | 0.826 | 0.917 |

Table 2: Chatterjee's correlation coefficient (average over all recordings) between EEG windows of varying length $w$ and the $x$-/$y$-coordinate of target (stimulus or gaze).

**Stem:** The stem is the initial part of the network responsible for converting the input signal into a form that can be processed by subsequent layers. In a traditional CNN architecture, the stem often includes a convolutional layer with a large kernel size, followed by a pooling layer to reduce the spatial dimensions of the input. For our FNN, we replaced the standard convolutional layer with a functional convolutional layer, using a large resolution to mimic the large kernel size of conventional CNN. In addition, we placed a spatial filtering layer in front of this layer, inspired by early experiments with the SpatialFilterCNN, where it was found to be beneficial for enhancing the model's performance. This design formed the foundation of the stem used in all the architectures we explored. The general structure of the stem is shown in Table 8 of Appendix A.1.

**Body:** The body of the network is where the bulk of the computation takes place. It is composed of multiple stages, each consisting of several blocks. For two-dimensional signals (like images) a typical block is structured as a sandwich of three convolutional layers, where the outer two use $1 \times 1$ kernels, and the middle layer uses a $3 \times 3$ kernel. In our architectures, we decided to omit the first $1 \times 1$ convolution following insights from the SpatialFilterCNN. Further, we set the kernel length of each filter to 9. Each convolutional layer was followed by batch normalization before applying the activation function, which is a common practice. Additionally, we incorporated residual connections within each block, allowing the input to be added to the output of the last convolutional layer. The resulting block structure is shown in Table 9 of Appendix A.1.

As in CNNs, each stage of our FNN body concludes with a pooling layer, that reduces the length of the input signal by a factor of two, effectively narrowing the signal as it progresses deeper into the network. Concurrently, the number of filters in the convolutional layers was increased at each stage, resulting in a deeper network.

**Head:** In a typical CNN, the head consists of one or more dense layers that transform the output of the body into the final prediction. To accommodate this, the output from the body must first be flattened or aggregated (*e.g.* through global average pooling).

Based on this reference architecture, we designed three FNNs and evaluated their performance. We did this by replacing different parts of the architecture with functional layers. An overview of the model architectures is provided in Figure 1 (b), while additional information are provided in the appendix. An implementation of the models under MIT license is available online: link to non-anonymous repository

## 5.1 Fully Functional Architecture

The first FNN follows a "fully functional" design, meaning that every component in the network is functional. For this, the residual block from the reference architecture is replaced by a "functional residual block", where the first convolution is changed to a functional convolutional layer. The second convolution, which uses a kernel size of 1, remains unchanged, as a functional convolutional layer with resolution 1 coincides with a standard convolutional layer. The functional residual block is displayed together with the standard residual block in Table 9 of Appendix A.1. Parts that differ between the two blocks are marked accordingly.

Multiple of such functional residual blocks are then chained together, making up one big stage, with the number of filters increasing progressively with the depth of the network. No pooling layers are used in or after the stage. There are two reasons for this: First, pooling operations with a stride break the smooth structure of the signals passing through the network. Second, using only functional layers at the head avoids the "parameter explosion" that would occur after the flattening operation.

In the head of the network, solely functional dense layers are employed. Because these layers expect functional inputs, there is no need to flatten the output of the body. Instead, the body's output is fed directly to the head. This leads to a significant reduction in the number of parameters. Without the need to flatten the body's output, the first dense layer in the head requires only `last_channels_out` × `neurons` weights. In contrast, if the output had been flattened, the number of weights would be `last_channels_out` × `last_steps` × `neurons`, where `last_steps` refers to the number of time steps remaining after the body.

For instance, with a window size of 512 and no pooling, the `last_steps` would be 512. Assuming `last_channels_out` is 256 and there are 256 neurons in the first dense layer, this requires only $256 \times 256 = 65,536$ weights, which equates to approximately $257\,\mathrm{KB}$, assuming 4-byte floats. In contrast, if the output had been flattened, the network would require $512 \times 256 \times 256 = 33,554,432$ weights, or approximately $128\,\mathrm{MB}$ with 4-byte floats.

The complete architecture of the first "fully functional" FNN is displayed in Table 10 of Appendix A.1.

## 5.2 Functional Body Architecture

The second FNN architecture takes a hybrid approach, using functional layers only in the body while concluding with standard dense layers in the head. Unlike the fully functional architecture, using standard dense layers in the head necessitates pooling layers at the end of each stage, to reduce the number of steps flowing into the head.

The body of this architecture is structured into two stages, each composed of two functional residual blocks. Pooling operations at the end of each stage reduce the input size by half, helping to control the dimensionality of the data as it passes through the network.

The output of the body is then flattened and passed through the head, which consisted of two standard dense layers. The first dense layer contains 64 neurons with an ELU activation function, while the second layer has two neurons with a linear activation function, producing the final output. Table 11 of Appendix A.1 provides a detailed breakdown of the second FNN architecture.

## 5.3 Minimally Functional Architecture

The third FNN architecture uses functional layers sparingly, with only a single functional dense layer in the head and the functional convolutional layer in the stem, that is part of all three architectures. The former functional layer helps avoid the need to flatten the body's output, which similar to the first architecture, reduces the number of required parameters. The "saved" parameters are reallocated to an additional larger

dense layer in the head, rather than adding more blocks to the body. This approach introduces some architectural variety. The body of this architecture is similar to that of the second FNN, but it employs standard residual blocks rather than functional ones. At the head, the output from the body is first aggregated by a functional dense layer with 512 neurons. This layer includes pooling and uses the ELU activation function. It acts as a more flexible global average pooling layer, providing the flexibility to weigh different parts of the signal differently, as opposed to standard global average pooling, which averages all inputs equally. After the functional layer, the data is processed by a standard dense layer with 512 neurons, also using the ELU activation function. The architecture concludes with a final standard dense layer with two neurons and a linear activation function for the output. The details of the third FNN architecture are shown in Table 12 of Appendix A.1.

## 6 Experiments

### 6.1 Metrics

Four metrics were used to evaluate model performance in the experiments: the Mean Euclidean Distance (MED) and RMSE, as described in Section 4.1, the Pearson correlation coefficient between true and predicted trajectories, and the precision. These metrics provide complementary insights into different aspects of model accuracy, helping us assess the quality of predictions across both smooth pursuit and saccadic eye movements.

Note that the term "precision", contrary to its use in classification, refers to a measure of variation of the predicted gaze position. More precisely, we define the precision as the MED between true and predicted direction of eye movement, i. e.,

$$\text{Precision} := \frac{1}{N} \sum_{i=1}^{N} \|\hat{y}_{i+1} - \hat{y}_i - (y_{i+1} - y_i)\|_2 \,, \tag{1}$$

where $y_i$ and $\hat{y}_i$ denote the true and predicted gaze position at time step $i$.

Since the Pearson correlation can only be computed for one-dimensional variables, we calculate the correlation separately for the $x$ and $y$ components of the predicted gaze positions. These separate metrics are referred to as $\text{corr}_x$ and $\text{corr}_y$, respectively. To obtain a single combined correlation metric, which can be used for tasks such as hyperparameter tuning, we compute the mean of these metrics.

While the MED serves as a primary measure of model performance, we introduced the Pearson correlation coefficient to complement the MED. It can, for example, detect models that predict constant values, such as the mean, as it will be (approximately) zero in this case.

Further, the Pearson correlation is translation and scale invariant. Models that predict generally the right direction, but not the correct position or scale, would be penalized by the MED, but will have a high correlation. This property makes the correlation especially valuable in early stages of model development, where capturing the direction is a positive sign of learning.

It is important to note, however, that while the correlation can provide valuable insights during model training, it is not as useful for benchmarking more advanced models. As models become more capable, they should not only capture the form but also accurately predict the scale and magnitude of the values. In these cases, MED becomes more relevant as a final evaluation metric, since it directly measures how close the predictions are to the true values.

### 6.2 Experimental Setup

The experiments were conducted on a Linux server, the hardware specifications of which are outlined in Table 3. The system features an AMD Ryzen Threadripper PRO 7995WX 96-Cores CPU running at up to 5.19 GHz, with 96 physical cores (192 threads). The server is equipped with 502 GiB (539 GB) of RAM. All experiments were run directly on the host operating system, which is Ubuntu 22.04.5 LTS. No GPUs were used in our setup.

| Component | Specification |
|---|---|
| CPU | AMD Ryzen Threadripper PRO 7995WX 96-Cores @ 5.19,GHz (96-Core/192 Threads) |
| RAM | 502,GiB DDR4 SDRAM |
| OS | Ubuntu 22.04.5 LTS |
| Python | 3.10.12 |

Table 3: Hardware specifications of the Linux server used for the experiments.

All experiments were conducted using Python 3.10.12. Hyperparameter tuning was handled using *Optuna*, with the TPE sampler configured for efficient search through the hyperparameter space.

## 7 Results and Discussion

### 7.1 Consumer-Grade EEG Eye Tracking

In the following, we compare the EEG-based reconstructions of eye movements with different baselines. The first baseline predicts the target's position randomly, and is used to verify if the model learns anything at all. As a stronger baseline, we use the mean position over the entire training set, which corresponds to the center of the screen. While this baseline is still trivial, it helps to check if the model learned something non-trivial. Finally, we use the webcam-based predictions as a strong baseline. This baseline is generally expected to perform well, up to some delay between movement and predictions, and is used for a direct comparison between EEG- and webcam-based eye tracking.

All models were trained on preprocessed data, as described in Appendix B.1.7-B.1.9. Missing values were imputed using a SARIMA model, followed by bandpass filtering (0.5 Hz-40 Hz) with notch filters at 50 Hz and 60 Hz. Input EEG windows of size 512 (2 seconds at 256 Hz) were standardized per channel using a standardization layer in the model's first layer. For training, windows were sampled with a stride of 4, while testing and validation used a stride of 1. The prediction target was the target's position at the last time instant in a given window.

Artifacts, such as muscle movements, were not explicitly filtered beyond bandpass filtering, as the baseline models were intentionally kept simple to provide reference points for future work.

We initially used the SpatialFilterCNN with its original hyperparameters, i.e., $N_S = 16$, $N_1 = 32$, $N_2 = 64$, `spatial_filtering` enabled, and `equally_sized` convolutional layers. While the original paper used a window size of 500, we opted for 512 samples to align with our sampling rate of 256 Hz, resulting in 2-second windows. Further, we used both filtered and unfiltered data. Details on the architecture of the SpatialFilterCNN are provided in Appendix A.2.

The results suggested that filtered data improves predictions for the "saccades" paradigm, while unfiltered data is better suited for smooth eye movements. Additionally, we conducted hyperparameter tuning experiments, applying the previously identified optimal filtering – filtered data for "saccades" and unfiltered data for "smooth" tasks. Using Optuna, the hyperparameters were tuned by training 20 models for 30 epochs with different hyperparameter configurations, for each of the four tasks. The ranges of the tuned hyperparameters are provided in Table 14 in the Appendix.

As additional baseline models, we used (i) an LSTM with two LSTM layers of 128 units and a dense layer with 2 neurons, and (ii) functional PCA (FPCA) combined with linear regression. In the context of sliding windows, the latter approach is expected to yield trivial results, since FPCA requires curve registration.

We used recordings of participants 1, 20, 28, 42, 52, and 69 as validation data. These recordings were selected as a representative subset from the training data. After hyperparameter tuning, the final model was trained on the entire training dataset.

The functional neural networks were trained exactly as the SpatialFilterCNN, i.e., by using a batch size of 384 for 30 epochs, early stopping at a patience of 5 epochs, and the Adam optimizer with a tuned learning

rate minimizing the mean-squared error. As before, we used Optuna to tune hyperparameters, as specified in Table 13.

The results of the level-1 and level-2 experiments are given in Tables 4 and 5, respectively. Additionally, exemplary predictions of different neural networks are displayed in Figure 2. For the functional neural networks, average results are reported; the corresponding standard deviations are displayed in Table 6.

Overall, all models made better predictions than random guessing. For level-1 experiments, the MED of the mean baseline is comparable or even lower than that of the webcam-based predictions. This effect is mainly due to the fact that level-1 experiments only included horizontal and vertical movements and short pauses in the center of the screen between these movements. The "mean baseline", remained at the origin, and predicted (at least) one coordinate correctly as 0, while the webcam-based predictions vary across both axes. When taking the correlations $\text{corr}_x$ and $\text{corr}_y$ into account, we see that the webcam-based predictions contain useful information about the target's position, while the mean baseline does not. Among the reference models, the combination of FPCA and linear regression led to (almost) trivial results that were only slightly better than the mean. The LSTM achieved good results for precision and, in smooth experiments, for the MED. However, considering the correlations, the predictions were only slightly better than the mean. Only the SpatialFilterCNN achieved meaningful results, with correlations around 0.2 and considerably low MED. The precision is relatively high, which indicates that successive predictions vary greatly and predicted trajectories are not continuous.

In general, the FNNs seem to fail predicting the target's $y$-coordinate, but outperform the SpatialFilterCNN in terms of $\text{corr}_x$. This is likely due to the experiment setup, since horizontal eye movements are more pronounced than vertical ones. For saccades experiments, the minimally functional and the functional body neural networks have an MED comparable to that of the SpatialFilterCNN. In case of smooth experiments, the MEDs of minimally and fully functional neural networks are slightly lower than that of the Spatial-FilterCNN. Notably, when comparing the SpatialFilterCNNs with the FNNs, the precision of the latter is substantially smaller. Note that small values of the precision, as defined in equation 1, are preferable. Low precision values indicate smoother predicted trajectories, and FNNs consistently achieve lower precision than the SpatialFilterCNN, indicating their ability to model coherent gaze paths.

For level-2 experiments, which introduce more degrees of freedom, the results look different. First, the "mean baseline" has a substantially higher MED, which stems from the fact that now two non-zero coordinates must be predicted. Overall, the webcam-based predictions yield the lowest MED and the highest correlations $\text{corr}_x$ and $\text{corr}_y$. The combination of FPCA and linear regression as well as the LSTM have results similar to the mean, indicating that they collapsed to constant models. For saccades experiments, the SpatialFilterCNN has a lower MED, yet a higher precision than the FNNs. For smooth experiments, FuncBody and MinFunc yield results similar to the SFCNN, in terms of MED and $\text{corr}_x$. Though, their precision is substantially lower, indicating smoother trajectories of predictions.

Generally, the SpatialFilterCNN has the lowest prediction error in terms of the MED. For smooth movements, the FNNs yield similar (level-2) or higher (level-1) correlations along the $x$-axis compared to the SpatialFilterCNN. This suggests that the FNNs are better at learning movement directions, while the SpatialFilterCNN is better at learning the exact magnitude of movements. The substantially lower precision metric of the FNNs suggests that their predictions are smoother and more consistent compared to those of the SpatialFilterCNN.

### 7.1.1 Ablation Study

In addition, we performed an ablation study, to evaluate the use of functional layers. For this, we used control models for each architecture, where the functional layers were replaced by conventional layers. More specifically, for the fully functional architecture, the functional residual blocks were replaced by standard residual blocks and the two final functional dense layers were substituted by two convolutional layers followed by a global average pooling layer. The first convolutional layer contained 256 filters with ELU activation, and the second contained 2 filters with a linear (no) activation. Both layers had a kernel size of 12. For the functional body, the functional residual blocks were replaced by standard residual blocks. Finally, for the minimally functional neural network, we replaced the single functional dense pooling layer with a standard

| level-1-saccades | | | level-1-smooth | | | |
| Model | MED | Precision | MED | Precision | $\text{corr}_x$ | $\text{corr}_y$ |
| --- | --- | --- | --- | --- | --- | --- |
| Random | 165.90 | 177.10 | 143.60 | 177.40 | $-0.01$ | 0.00 |
| Mean | 83.58 | **0.65** | **51.98** | **0.36** | 0.00 | 0.00 |
| Webcam | **81.89** | 1.48 | 74.59 | 0.89 | **0.69** | **0.35** |
| SFCNN | **54.62** | 4.65 | 57.92 | 6.09 | **0.20** | **0.19** |
| FPCA | 89.71 | 2.25 | 50.23 | 0.75 | 0.04 | 0.05 |
| LSTM | 83.35 | **0.64** | **43.42** | **0.37** | 0.00 | 0.00 |
| FullyFunc | 61.82 | **1.71** | 56.02 | 1.18 | 0.21 | 0.01 |
| FuncBody | 56.76 | 1.86 | 67.18 | **1.15** | 0.17 | **0.03** |
| MinFunc | **55.82** | 1.83 | **55.53** | 1.25 | **0.23** | 0.01 |

Table 4: Results of the baseline and reference models for level-1 experiments. Functional neural networks were trained 5 times and average values are reported. The best result for each metric in each category is marked in bold.

| level-2-saccades | | | level-2-smooth | | | |
| Model | MED | Precision | MED | Precision | $\text{corr}_x$ | $\text{corr}_y$ |
| --- | --- | --- | --- | --- | --- | --- |
| Random | 199.80 | 177.30 | 164.50 | 177.50 | 0.00 | 0.00 |
| Mean | 162.30 | **0.59** | 104.50 | **0.42** | 0.00 | 0.00 |
| Webcam | **87.19** | 1.47 | **77.00** | 1.00 | **0.90** | **0.75** |
| SFCNN | **109.00** | 6.45 | **99.38** | 20.84 | **0.43** | **0.16** |
| FPCA | 161.83 | 2.34 | 104.62 | **0.57** | 0.00 | 0.00 |
| LSTM | 162.85 | **0.61** | 104.14 | **0.43** | 0.25 | $-0.02$ |
| FullyFunc | 123.47 | 2.55 | 111.17 | 1.67 | 0.36 | **0.13** |
| FuncBody | 123.11 | 3.25 | **99.33** | 1.15 | **0.43** | 0.11 |
| MinFunc | **120.93** | **2.00** | 99.78 | **1.04** | **0.43** | 0.08 |

Table 5: Results of the baseline and reference models for level-2 experiments. Functional neural networks were trained 5 times and average values are reported. The best result for each metric in each category is marked in bold.

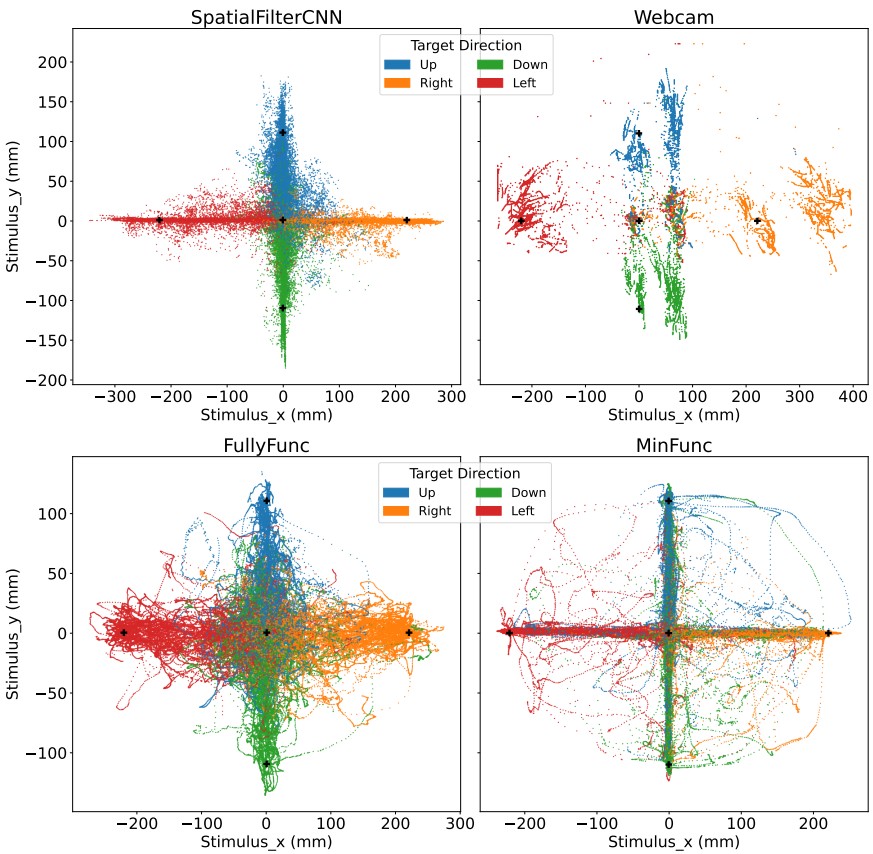

Figure 2: Comparison of various neural networks on the level-1 saccades task. Every prediction made on the test set is shown as a dot. The dots are colored based on the true target position. The four target positions are indicated by black crosses.

| Model | MED | Precision | RMSE | $\mathrm{corr}_x$ | $\mathrm{corr}_y$ |
|---|---|---|---|---|---|
| *Panel A: level-1-saccades* | | | | | |
| FullyFunc | 61.82 (±1.41) | **1.71** (±0.05) | **87.64** (±0.68) | - | - |
| FullyFunc (control) | **56.77** (±0.97) | 3.16 (±0.07) | 90.18 (±0.97) | - | - |
| FuncBody | **56.76** (±1.95) | **1.86** (±0.18) | **85.89** (±1.21) | - | - |
| FuncBody (control) | 63.41 (±0.89) | 2.12 (±0.04) | 88.64 (±0.71) | - | - |
| MinFunc | **55.82** (±1.06) | 1.83 (±0.30) | **84.64** (±2.14) | - | - |
| MinFunc (control) | 73.68 (±15.72) | **1.40** (±1.06) | 109.87 (±19.52) | - | - |
| *Panel B: level-1-smooth* | | | | | |
| FullyFunc | 56.02 (±2.93) | **1.18** (±0.30) | 81.12 (±3.25) | **0.21** (±0.06) | 0.01 (±0.03) |
| FullyFunc (control) | **53.73** (±3.48) | 1.25 (±0.81) | **78.60** (±2.34) | 0.14 (±0.13) | **0.02** (±0.03) |
| FuncBody | 67.18 (±3.86) | **1.15** (±0.18) | 91.74 (±5.23) | 0.17 (±0.04) | 0.03 (±0.03) |
| FuncBody (control) | **57.52** (±0.96) | 1.44 (±0.04) | **82.60** (±1.56) | **0.23** (±0.03) | **0.04** (±0.02) |
| MinFunc | 55.53 (±2.78) | 1.25 (±0.48) | 81.78 (±3.45) | **0.23** (±0.01) | **0.01** (±0.02) |
| MinFunc (control) | **52.08** (±1.80) | **0.36** (±0.00) | **76.82** (±0.59) | 0.00 (±0.00) | 0.00 (±0.00) |
| *Panel C: level-2-saccades* | | | | | |
| FullyFunc | **123.47** (±1.70) | 2.55 (±0.41) | **143.81** (±2.12) | - | - |
| FullyFunc (control) | 163.58 (±0.30) | **0.60** (±0.00) | 177.49 (±0.20) | - | - |
| FuncBody | **123.11** (±2.69) | 3.25 (±0.95) | **143.41** (±4.93) | - | - |
| FuncBody (control) | 163.18 (±0.04) | **0.60** (±0.00) | 177.25 (±0.01) | - | - |
| MinFunc | **120.93** (±0.97) | **2.00** (±0.22) | **140.65** (±1.80) | - | - |
| MinFunc (control) | 132.45 (±1.83) | 3.61 (±0.06) | 157.07 (±1.92) | - | - |
| *Panel D: level-2-smooth* | | | | | |
| FullyFunc | 111.17 (±0.91) | **1.67** (±0.03) | 128.15 (±1.19) | 0.36 (±0.03) | **0.13** (±0.04) |
| FullyFunc (control) | **103.58** (±0.59) | 1.89 (±0.08) | **118.15** (±0.68) | **0.41** (±0.01) | 0.11 (±0.02) |
| FuncBody | **99.33** (±0.67) | 1.15 (±0.19) | **113.36** (±0.91) | **0.43** (±0.01) | **0.11** (±0.02) |
| FuncBody (control) | 104.61 (±0.01) | **0.42** (±0.00) | 121.46 (±0.00) | 0.00 (±0.00) | 0.00 (±0.00) |
| MinFunc | **99.78** (±1.82) | 1.04 (±0.14) | **113.67** (±2.22) | **0.43** (±0.03) | **0.08** (±0.06) |
| MinFunc (control) | 104.56 (±0.00) | **0.42** (±0.00) | 121.45 (±0.00) | 0.00 (±0.00) | 0.00 (±0.00) |

Table 6: Results of the functional and control models. The better result between functional and control model is highlighted in bold, and the best result for each metric is underlined.

convolutional layer, a global average pooling layer, and an ELU activation. The convolutional layer had 512 filters, to match the 512 neurons of the functional dense layer, and a kernel size of 12. Global average pooling was used to obtain a scalar output, corresponding to the output of the functional dense pooling layer.

The results are displayed in Table 6. For some model-task combinations, the functional version yields better results, while for other combinations the conventional version seems preferrable. Excluding level-2-smooth experiments, MinFunc models generally seem to produce better predictions. The functional variant performs best on saccade experiments, whereas the control model seems superior on smooth experiments. In the level-2-smooth condition, the functional FuncBody achieves the lowest MED and RMSE. However, note that $\mathrm{corr}_x$ is 0 for the MinFunc control model, which indicates that the model collapsed and yields trivial predictions. Accounting for this effect, the functional MinFunc yields the overall best results, indicating a benefit of the functional approach.

## 7.2 EEGEyeNet

The previous comparison is based on data recorded by consumer-grade hardware. In addition, we compare FNNs with conventional CNNs based on the EEGEyeNet dataset, which was recorded under laboratory-controlled conditions and with research-grade EEG equipment with more electrodes and wet sensors. This allows an evaluation of FNNs under optimal conditions, compared to the real-world conditions from the previous section.

| Model | MED | MAE |
|---|---|---|
| FullyFunc | **68.5** ($\pm 1.0$) | **42.7** ($\pm 0.6$) |
| FullyFunc (control) | 69.9 ($\pm 1.3$) | 43.6 ($\pm 0.8$) |
| FuncBody | **68.0** ($\pm 0.8$) | 42.4 ($\pm 0.6$) |
| FuncBody (control) | 68.1 ($\pm 0.5$) | **42.3** ($\pm 0.3$) |
| MinFunc | 66.8 ($\pm 0.5$) | 41.5 ($\pm 0.4$) |
| MinFunc (control) | **66.2** ($\pm 0.8$) | **41.1** ($\pm 0.5$) |
| EEGNet | 77.3 ($\pm 0.3$) | 48.7 ($\pm 0.2$) |
| SpatialFilterCNN | 68.8 ($\pm 1.4$) | 42.9 ($\pm 0.8$) |

Table 7: Results of various models on the EEGEyeNet dataset. The mean (and standard deviation) of the MED and MAE are shown for each model. The better result between the functional and control model is highlighted in bold, and the best result for each metric is underlined. The MED and MAE are reported in millimeters using a conversion factor of 0.5.

We used the same architectures and hyperparameters as before. Furthermore, we have added the EEGNet, a standard neural network for the analysis of EEG data, to the comparisons.

For training, the Adam optimizer with a learning rate of 0.0001 and a batch size of 64 was used. Further, we employed early stopping based on the validation loss, with patience of 20 epochs. All models were trained for a maximum of 50 epochs, with a window size of 500, as specified by Kastrati et al. (2021). In line with the literature, each model was trained 5 times. The results, including the mean and standard deviation of the MED and mean absolute error (MAE), are summarized in Table 7.

Except for the control model of the fully functional neural network, all functional architectures outperformed the SpatialFilterCNN in both the MED and MAE metrics, resulting in new state-of-the-art results on the EEGEyeNet benchmark. The best-performing model was the MinFunc control model, which surpassed the SpatialFilterCNN by 2.6 mm in the MED and 1.8 mm in the MAE.

It is important to note that the architectures used in these experiments were not tuned specifically for the EEGEyeNet dataset, and it is likely that further improvements could be achieved with hyperparameter tuning. Unlike the results from Section 7.1, where functional layers showed a clear benefit, the differences between the functional, and control models on the EEGEyeNet dataset were much smaller. In fact, the control models outperformed the functional models half of the time. Therefore, the advantage of functional layers in the constructed architectures is less clear when evaluated on the EEGEyeNet dataset.

Finally, we note that surpassing the current state of the art with a model architecture largely based on standard convolutional network architectures suggests that there is still significant room for improvement in the field of EEG-based eye tracking.

## 8 Conclusion

We introduced the "Consumer-Grade EEG-based Eye Tracking" dataset as a new benchmark for methods in the field of functional data analysis. Additionally, we stated a number of open challenges (classification, clustering, dimension reduction, outlier and change point detection) related to the dataset. We further studied the use of functional neural networks to solve the main challenge associated with the dataset, reconstructing a target's position on the screen from raw EEG data. This task can be formulated as function-on-function or scalar-on-function regression problem. We provided benchmark results for the latter. While our focus was on the main challenge, the remaining tasks outlined, such as classification and change point detection, offer valuable opportunities for further exploration. Baseline results for these tasks are yet to be established and provide a promising direction for future research.

While results on the EEGEyeNet dataset were less conclusive, findings in Section 7.1 indicate that functional neural networks have beneficial properties for the prediction of the target's position. Across all paradigms, the MinFunc model achieved an MED comparable to the SpatialFilterCNN, but with a substantially better

(lower) precision. This lower precision likely reflects the tendency of FNN predictions to be smoother, as illustrated in Figure 2. In the ablation study comparing functional and conventional neural networks, functional MinFunc model generally produced the best results. Overall, these results suggest that FNNs are well suited for predicting smooth targets, and are particularly promising for analyzing functional data.

In conclusion, the "Consumer-Grade EEG-based Eye Tracking" dataset allows a comparison of FDA methods across different open challenges. While FNNs have been found to be a helpful method for analyzing the dataset, it remains an open question how well other approaches work. It is expected that progress on this dataset will advance both EEG analysis and functional data analysis.

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

# A Details on Model Architectures

## A.1 Functional Neural Networks

| Layer | Parameters | Output Shape |
|---|---|---|
| Input | — | $(\text{batch\_size}, \text{window\_size}, 4)$ |
| Conv1D | kernel_size: 1, filters: 16 | $(\text{batch\_size}, \text{window\_size}, 16)$ |
| BatchNorm | axis: -1 | $(\text{batch\_size}, \text{window\_size}, 16)$ |
| ReLU | — | $(\text{batch\_size}, \text{window\_size}, 16)$ |
| FuncConv1D | padding: same, resolution: 128, n_functions: 9, basis_type: Fourier | $(\text{batch\_size}, \text{window\_size}, 64)$ |
| AvgPool | pool_size: 2, strides: 2 | $(\text{batch\_size}, \text{window\_size}/2, 64)$ |

Table 8: The general structure of the stem, which is used by all three FNN architectures.

| Layer | Parameters | Output Shape |
|---|---|---|
| Input | — | $(\text{batch\_size}, \text{steps}, \text{channels\_in})$ |
| (Func)Conv1D | padding: same, filters: channels_out (*standard only*) kernel_size: 9 (*functional only*) resolution: 24, n_functions: 6, basis_type: Legendre | $(\text{batch\_size}, \text{steps}, \text{channels\_out})$ |
| BatchNorm | axis: -1 | $(\text{batch\_size}, \text{steps}, \text{channels\_out})$ |
| elu | — | $(\text{batch\_size}, \text{steps}, \text{channels\_out})$ |
| Conv1D | padding: same, filters: channels_out, kernel_size: 1 | $(\text{batch\_size}, \text{steps}, \text{channels\_out})$ |
| BatchNorm | axis: -1 | $(\text{batch\_size}, \text{steps}, \text{channels\_out})$ |
| Add | — | $(\text{batch\_size}, \text{steps}, \text{channels\_out})$ |
| elu | — | $(\text{batch\_size}, \text{steps}, \text{channels\_out})$ |

Table 9: The structure of a standard ResBlock and a FuncResBlock.

**Architecture #1: Fully Functional**

| Layer | Parameters | Output Shape |
|---|---|---|
| STEM | — | $(\text{batch\_size}, \text{window\_size}/2, 64)$ |
| FuncResBlock | $N_1$ filters | $(\text{batch\_size}, \text{window\_size}/2, 64)$ |
| FuncResBlock | $N_2$ filters | $(\text{batch\_size}, \text{window\_size}/2, 96)$ |
| FuncResBlock | $N_3$ filters | $(\text{batch\_size}, \text{window\_size}/2, 144)$ |
| FuncResBlock | $N_4$ filters | $(\text{batch\_size}, \text{window\_size}/2, 216)$ |
| FuncDense | n_functions, basis_type, neurons: 256, activation: elu | $(\text{batch\_size}, \text{window\_size}/2, 256)$ |
| FuncDense | n_functions, basis_type, neurons: 2, activation: linear, pooling: True | $(\text{batch\_size}, 2)$ |

Table 10: Architecture of the "fully functional" neural network. Parameters without specified values are tuned.

**Architecture #2: Functional Body**

| Layer | Parameters | Output Shape |
|---|---|---|
| STEM | — | $(\text{batch\_size}, \text{window\_size}/2, 64)$ |
| FuncResBlock | $N_1$ filters | $(\text{batch\_size}, \text{window\_size}/2, 64)$ |
| FuncResBlock | $N_1$ filters | $(\text{batch\_size}, \text{window\_size}/2, 64)$ |
| AvgPool | pool_size: 2, strides: 2 | $(\text{batch\_size}, \text{window\_size}/4, 64)$ |
| FuncResBlock | $N_2$ filters | $(\text{batch\_size}, \text{window\_size}/4, 112)$ |
| FuncResBlock | $N_2$ filters | $(\text{batch\_size}, \text{window\_size}/4, 112)$ |
| AvgPool | pool_size: 2, strides: 2 | $(\text{batch\_size}, \text{window\_size}/8, 112)$ |
| Flatten | — | $(\text{batch\_size}, \text{window\_size}/8 \times 112)$ |
| Dense | neurons: 64, activation: elu | $(\text{batch\_size}, 64)$ |
| Dense | neurons: 2, activation: linear | $(\text{batch\_size}, 2)$ |

Table 11: Architecture of the "functional body" neural network. Parameters without specified values are tuned.

## A.2   SpatialFilterCNN

A spatial filter, in contrast to temporal filters, is a filter that acts across different electrodes, combining signals at a single time point. Fuhl et al. (2023) proposed a neural network architecture, where the first layer shall act as a (learnable) spatial filter, for the EEGEyeNet data. We refer to this model as *SpatialFilterCNN*.

The model consists of a spatial filtering layer, two residual blocks and two fully connected layers at the output. The detailed architecture of the SpatialFilterCNN model is shown in Figure 3. In case of the EEGEyeNet data, the input is a matrix of shape $500 \times 128$ (time points $\times$ channels). The spatial filtering layer is a one-dimensional convolutional layer with 16 filters and a kernel size of 1. It is followed by a batch normalization layer and a ReLU activation function. The output of this layer is then passed through two residual blocks. Each residual block consists of two one-dimensional convolutional layers, with either 32 filters for the first residual block or 64 filters for the second. The first convolutional layer in each residual

**Architecture #3: Minimally Functional**

| Layer | Parameters | Output Shape |
|---|---|---|
| STEM | — | $(\text{batch\_size}, \text{window\_size}/2, 64)$ |
| ResBlock | $N_1$ filters | $(\text{batch\_size}, \text{window\_size}/2, 64)$ |
| ResBlock | $N_1$ filters | $(\text{batch\_size}, \text{window\_size}/2, 64)$ |
| AvgPool | pool_size: 2, strides: 2 | $(\text{batch\_size}, \text{window\_size}/4, 64)$ |
| ResBlock | $N_2$ filters | $(\text{batch\_size}, \text{window\_size}/4, 112)$ |
| ResBlock | $N_2$ filters | $(\text{batch\_size}, \text{window\_size}/4, 112)$ |
| AvgPool | pool_size: 2, strides: 2 | $(\text{batch\_size}, \text{window\_size}/8, 112)$ |
| FuncDense | n_functions, basis_type, neurons: 512, activation: elu, pooling: True | $(\text{batch\_size}, 512)$ |
| Dense | neurons: 512, activation: elu | $(\text{batch\_size}, 512)$ |
| Dense | neurons: 2, activation: linear | $(\text{batch\_size}, 2)$ |

Table 12: Architecture of the "minimally functional" neural network. Parameters without specified values are tuned.

| Hyperparameter | Range / Setting |
|---|---|
| Temporal filtering | Enabled or Disabled |
| Learning rate | $10^{-5}$ to $10^{-1}$ |
| $N_1$ (number of filters) | 4 to 64 |
| $N_2$ (number of filters) | 8 to 128 |
| $N_3$* (number of filters) | 16 to 256 |
| $N_4$* (number of filters) | 32 to 512 |
| Basis type | Legendre or Fourier |
| Resolution | 12 to 256 |
| Number of basis functions | 6 to 128 |

Table 13: Hyperparameter ranges of the functional neural networks. Hyperparameters marked with "*" correspond to the fully functional neural network.

block has a kernel size of 9, and uses padding to keep the first dimension of the output shape the same as the input shape. This is followed by a Batch Normalization layer and a ReLU activation function.

The second convolutional layer has a kernel size of 1 and is followed by another Batch Normalization layer. The output of the second convolutional layer is then added to the input of the residual block, which is passed through another one-dimensional convolutional layer in order to match the output shape of the second convolutional layer. The added output is then passed through a ReLU activation function and an average pooling layer with a pool size of 2 and a stride of 2. After the second residual block the output is flattened and passed through a fully connected layer with 256 neurons and a ReLU activation function. The output of this layer is then passed through another fully connected layer with 2 neurons, which is the output of the model and represents the prediction of the gaze position in pixels.

The hyperparameters of the SpatialFilterCNN are given in Table 14.

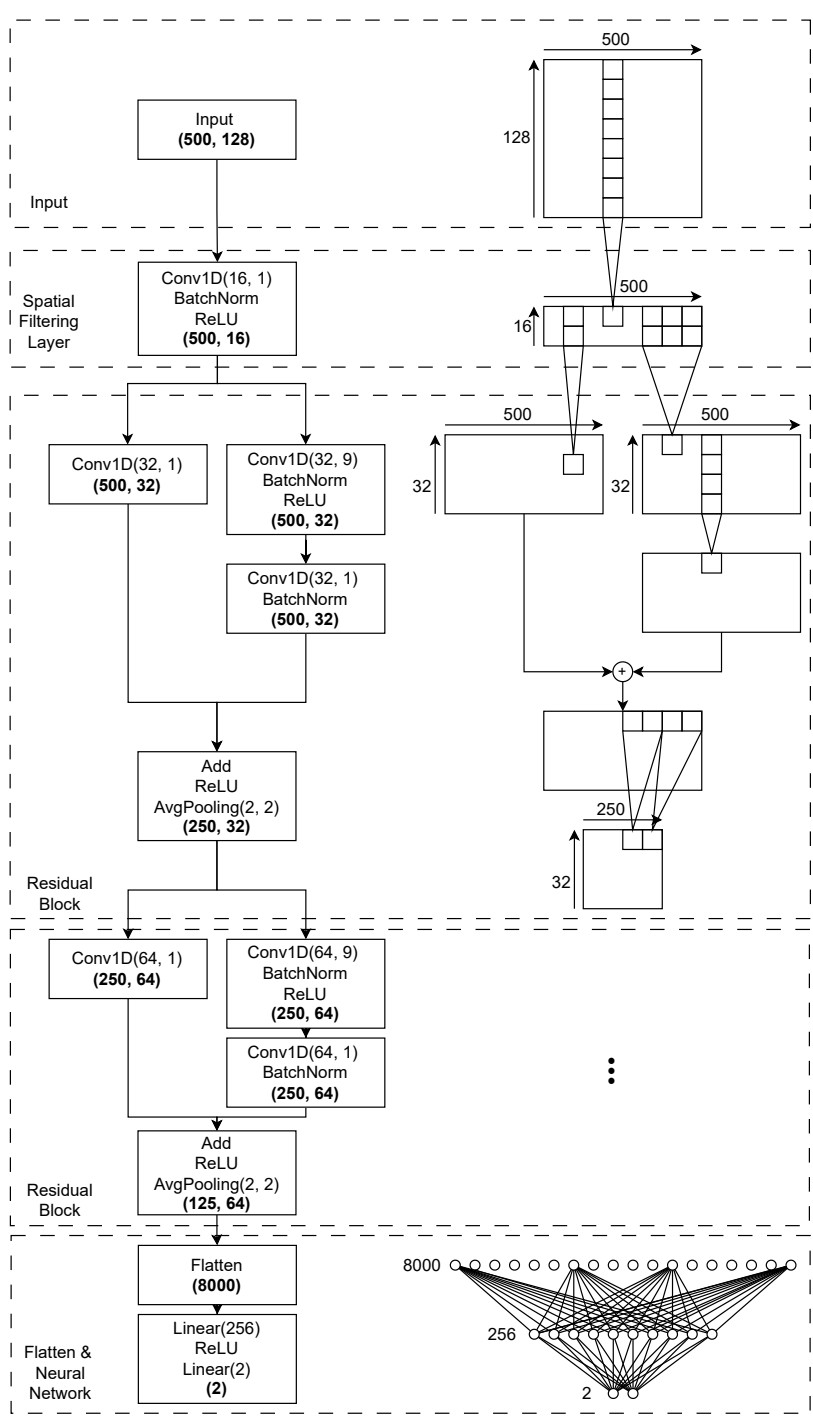

Figure 3: The architecture of the SpatialFilterCNN model, which on a high level consists of the input, a spatial filtering layer, two residual blocks and a fully connected neural network at the output. Conv1D($n$, $m$) denotes a one-dimensional convolutional layer with $n$ filters and a kernel size of $m$, AvgPooling($n$, $m$) average pooling with pool size $n$ and stride $m$ and Linear($n$) a layer of $n$ fully connected neurons.

| Hyperparameter | Range / Setting |
|---|---|
| Window size | 128 to 1024 samples |
| Learning rate | $10^{-5}$ to $10^{-1}$ |
| Spatial filtering | Enabled or Disabled |
| Equally sized kernels | Enabled or Disabled |
| $N_S$ (number of spatial filters) | 4 to 64 |
| $N_1$ (filters in first residual block) | 8 to 128 |
| $N_2$ (filters in second residual block) | 16 to 256 |

Table 14: Hyperparameter ranges of the SpatialFilterCNN.

## B  Data Description

### B.1  Experiments

The dataset consists of simultaneously recorded EEG and eye tracking data from 113 participants, collected during 116 sessions, in which 4 experimental paradigms were presented to the participants. The experimental paradigms are designed in a way that it becomes increasingly challenging to reconstruct the gaze position from the EEG data as the eye movement becomes less restricted. In every paradigm, the participants were instructed to follow a target moving on the screen as accurately as possible with their eyes. The presentation was also designed in a way that helped the participants follow the target closely.

### B.1.1  Participants

A total of 113 participants (91 males and 22 females, 100 right-handed and 12 left-handed, 1 ambidextrous) took part in this study. Demographic data, including the age, gender, handedness, vision correction, neurological disorders, and color blindness of the participants, is summarized in Table 15, the age distribution is shown in Figure 4. The participants were informed about the purpose of the study, the data that would be collected, the duration of the study, and that any collected data would be anonymized. All participants took part in this study voluntarily and received no compensation.

In accordance with the guidelines of the German Research Foundation (DFG), no ethics vote was required for the existing study (German Research Foundation, 2023). More specifically, the DFG states that "A statement by an ethics committee is not usually required for studies involving electrophysiological leads (e.g. EEG, MEG, NIRS) if the above points do not apply.", where the "above points" refer to experiments involving high levels of stress, patients, (f)MRI or psychopharmacological studies; none of which applies here. In accordance with the GDPR, the EEG recordings were anonymized so that no conclusions can be drawn about the study participants.

Every participant was asked to sign two documents. The first document was a consent form, where the participants agreed to take part in the study, and the data collection. Additionally, participants confirmed that they were taking part in the study voluntarily and were not receiving any compensation. The second document was the demographic questionnaire, where the participants provided information about their age, gender, handedness, vision correction, neurological disorders, and color blindness.

The participant pool shows a noticeable demographic imbalance, with 81% male and 88% right-handed individuals (Table 15). While these proportions reflect the voluntary nature of recruitment within the given university course setting, they may introduce bias in downstream analyses, if ocular artifacts or EEG signal characteristics vary systematically by gender or handedness. This limitation should be considered when interpreting results derived from the dataset.

### B.1.2  Setup

The participants were seated in front of a 24 inch desktop monitor with a resolution of $2560\,\mathrm{px} \times 1440\,\mathrm{px}$ and a 60 Hz refresh rate, placed 60 cm away from the participants and raised high enough such that the center of

| Demographic | No. of Participant | Percentage |
|---|---|---|
| Gender (male/female/diverse) | 91 / 22 / 0 | 81% / 19% / 0% |
| Handedness (left/right/ambidextrous) | 12 / 100 / 1 | 11% / 88% / 1% |
| Vision Correction (yes/no) | 54 / 59 | 48% / 52% |
| Neurological Disorder (yes/no) | 6 / 107 | 5% / 95% |
| Color Blind (yes/no) | 2 / 111 | 2% / 98% |

Table 15: Demographic data of the participants.

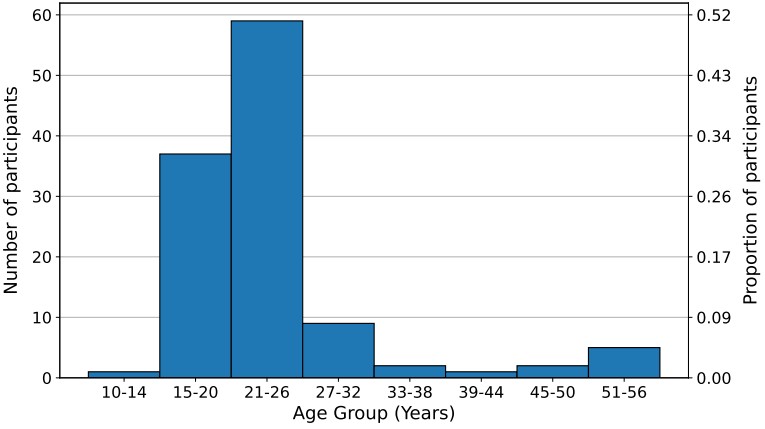

Figure 4: Histogram of the age distribution of the participants. The left y-axis shows the number of participants in each age group, the right y-axis shows the proportion of participants in each age group.

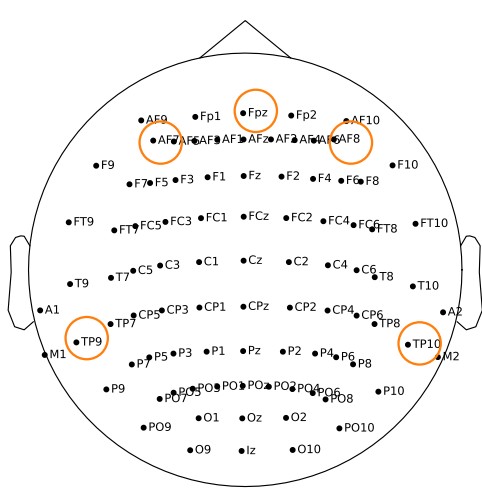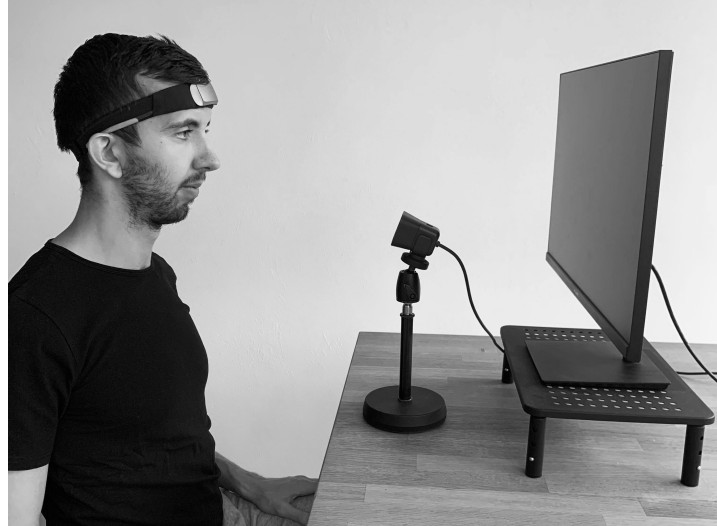

Figure 5: Left: Positions of electrodes of the Muse S 2 Headband in the 10-20 system. Right: Recording setup of the experiments. Consent was obtained for publication of the individual's facial image.

the screen matched the eye height of the subject. A webcam was mounted on a stand in front of the screen, adjusted to be as high as the lower edge of the screen, looking up at the subject. The recording setup is illustrated on the right of Figure 5.

### B.1.3   Hardware

The webcam used was the Logitech StreamCam, which records in Full HD (1920 px×1080 px), at 60 frames per second. The monitor used was a DELL P2416D with a 2560x1440 resolution, 59.95Hz refresh rate, and 8-bit RGB Standard Dynamic Range. The EEG data was recorded using the Muse S 2 Headband (RRID: SCR_014418), a consumer-grade EEG-Headset with five dry electrodes TP9, TP10, AF7, AF8 and Fpz, where the latter is used as reference electrode. The headband records at a sampling rate of 256 Hz. The electrode placement in the standard 10-20 system is shown on the left of Figure 5.

Two laptops were used: one for controlling the stimuli presentation and recording the data and one for showing the stimuli to the participants. Both laptops communicated via a socket connection.

No additional calibration beyond the default manufacturer procedures was performed for either device type. The Logitech StreamCam operated with factory-set optical parameters such as focal length (3.7 mm), aperture size (f/2.0) and field of view (78° diagonal) Logitech (2025). Lighting conditions during sessions were not measured in lux but corresponded to typical indoor classroom illumination. For the Muse S 2 headband, electrode impedance thresholds were not recorded. Signal quality was assessed visually using the provided software. Detailed hardware calibration parameters were not systematically documented, reflecting the practical constraints and real-world focus of this consumer-grade dataset.

### B.1.4   Software

Data from three different sources was recorded: the position of the target, the gaze position, and the EEG data. The resulting three data streams were synchronized using the Lab Streaming Layer Protocol (LSL). LSL is an open-source framework widely used for multimodal time-synchronized physiological signal acquisition Kothe et al. (2025). The LSL streams were recorded using the LabRecorder software, which properly resolves the synchronization information of multiple streams. The gaze position was estimated from the webcam video using the GazePointer software.

| Experimental Paradigm | Duration | Total Duration |
|---|---|---|
| level-1-smooth | 56 s | 1 h 49 min |
| level-1-saccades | 1 min | 1 h 56 min |
| level-2-smooth | 2 min 8 s | 4 h 8 min |
| level-2-saccades | 2 min | 3 h 52 min |
| Total | 6 min 4 s | 11 h 45 min |

Table 16: Duration of the experimental paradigms.

### B.1.5   Experiment Procedure

Most sessions took place during 90 minute practice groups of different artificial intelligence related courses at the Darmstadt University of Applied Sciences. The participants were seated in the back of the room in which the practice group took place. At the start of the group exercise, before any session started, all participants were briefed about the study and the data collection. The participants were informed about the purpose of the study, the data that would be collected, the duration of the study, that participation was voluntary, and that any collected data would be anonymized. After the briefing, the first subject was seated in front of the screen and the first session started.

Each session (nominally) lasted ten minutes in total and consisted of the following parts: introduction, practice, and recording. The introduction included the calibration of the camera-based eye tracker, fitting the EEG headset to the participant's head, and checking signal quality. Furthermore, the distance to the screen and the height of the screen were adjusted to match the eye height of the subject. The participants were instructed not to move or speak during the recording. Before each recording, a short practice session was conducted to prepare the participants for the respective experimental paradigm.

During the actual data collection, EEG and eye tracking data were recorded in four different situations: level-1-smooth, level-1-saccades, level-2-smooth, and level-2-saccades. The duration of each paradigm is shown in Table 16 and was chosen to balance sufficient data collection per condition with participant comfort and to minimize fatigue effects observed during pilot trials. A level-1 experiment only required the subject to move their eyes left, right, up or down for one minute. Data from those experiments is meant to serve as a first stepping stone for potential EEG-based eye tracking techniques.

In a "smooth" experiment, the target moved across the screen in a continuous manner. The subject followed the target using eye movement called "smooth pursuit", in which the eyes move smoothly to follow a moving object. A "saccades" experiment required the subject to follow the target using a series of quick eye movements called "saccades". Here, the target jumped between predefined points on the screen. For all paradigms, the stimuli presentation was designed in such a way, that the user could always anticipate the movement or a jump of the target. This way, the position of the target is always close to the ground truth gaze position. If the participant experienced eye strain or fatigue, a break was taken before the next experiment.

### B.1.6   Stimulus Presentation

The target was presented to the participants using a custom stimuli presentation software ("Presentation App") written in Python using the Qt framework. The bounding box in all tasks was 440 mm x 220 mm and the target was presented as a circle with a diameter of 10 mm.

**Saccade Experiments.** The grid point positions in the level-1 and level-2 "saccades" experiments are shown in Figure 6 (left). In a level-1 experiment, there was a saccade to the edge of the bounding box every second, followed by a saccade back to the center of the screen in the next second. In a level-2 experiment, the target jumped to any of the grid points every 1.5 seconds.

**Smooth Experiments.** The path along which the target moved in the "smooth" experiments was created using a parameterized curve. More specifically, for a given $t \in [0, 1]$, the $x$ and $y$ position of the target in

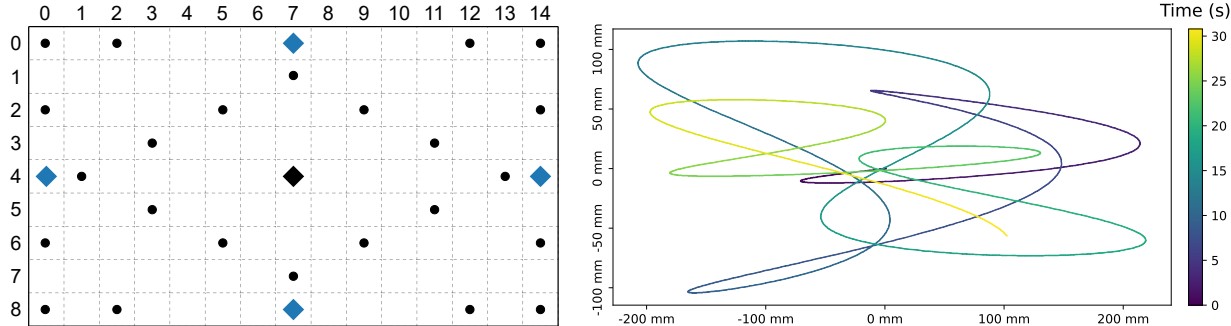

Figure 6: Left: Grid points in the "saccades" experiments. Displayed as blue diamonds (level-1 experiment), black dots (level-2 experiment) and black diamond (both). Right: Exemplary trajectory of the target that was presented to the participants. It was created using the parameterized function equation 2.

millimeters relative to the center of the screen was given by:

$$f : [0, 1] \to \mathbb{R}^2, \quad t \mapsto \begin{pmatrix} (\cos(at)\sin(bt) + et) \cdot \frac{w}{2} \\ (\cos(ct)\sin(dt) + ft) \cdot \frac{h}{2} \end{pmatrix}, \tag{2}$$

where $w$ and $h$ are the width and height of the bounding box in millimeters, i.e., $w = 440\,\text{mm}$ and $h = 220\,\text{mm}$. In particular, the $x$-coordinate describes the horizontal and the $y$-coordinate the vertical position of the target. The time in seconds the target takes to traverse the complete curve can be controlled by the parameter $T$. The value of $t$ is incremented 120 times per second, in steps of size $\frac{1}{120}/T$, resulting in a smooth movement.

For all level-1 experiments $a = b = c = d = 0$, $e, f \in \{-1, 0, 1\}$. This results in paths where the target moves continuously from the center to one of the edges of the specified bounding box and back. For these experiments, $T = 1$, for vertical movements and $T = 2$ for horizontal movements. This choice resulted in the target taking 1 second for vertical and 2 seconds for horizontal movements. The different times were chosen to account for the different height and width of the bounding box.

For level-2 experiments $a, b, c, d$ were randomly selected from the interval $[-50, 50]$, $e = f = 0$ and $T = 28.5$, so that the target takes 28.5 seconds to traverse an entire curve. The random selection of parameters results in more complex paths on which the target moves. An example of a path created using this method is displayed in Figure 6 (right). Multiple curves were shown in each experiment. Before the target started to move along the next curve, it waited for 2 seconds in the center of the screen. The value of $T$ was selected so that completing four curves and the three intervening pauses took a total of 2 minutes.

In a level-1 smooth experiment, only four possible curves existed, one for each direction. The order in which the curves were shown was determined by randomly sampling without replacement from a pool where every direction existed 4 times, resulting in a sequence of 16 curves in total. In level-2 smooth experiments, four curves were shown which were created by randomly picking values for $a, b, c, d$ from the interval $[-50, 50]$. In total, 18 different of these curves were created.

In level-1 saccades experiments, the target jumped 30 times to one of the four directions (left, right, up, down) and back to the center, resulting in 60 saccades in total. To choose the 30 directions for the jumps, 8 random permutations of the four directions were created, and the last two directions from the final permutation were discarded. For the level-2 saccades experiments, 80 positions were selected by creating 4 random permutations of the 25 possible positions. The last 20 positions from the final sequence were then discarded.

Several design choices were made to help the participants anticipate the target's movement, to minimize the distance between the position of the target and the participant's gaze. The presentation differed between the "smooth" and "saccades" paradigms. In the "smooth" experiments, in addition to the yellow circle moving across the screen as the target, a line was displayed to show the expected direction of the target's movement.

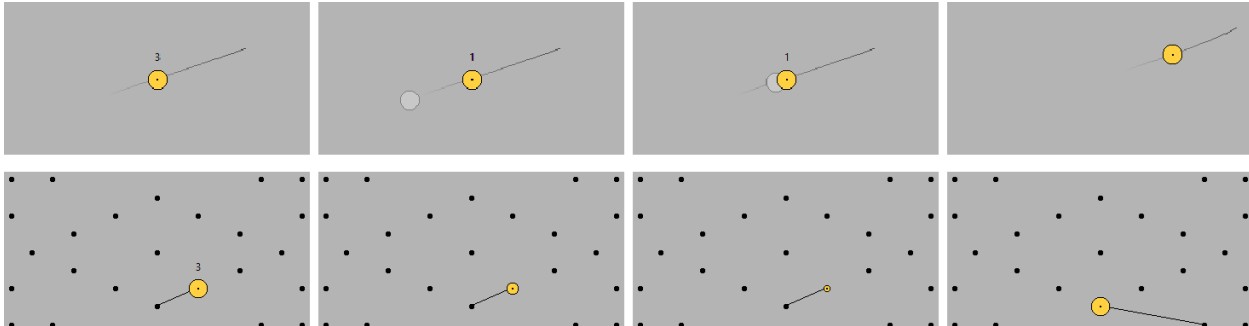

Figure 7: Stimuli presentation. Top: Presentation of the stimulus in the level-2 smooth experiment (countdown at the start - the "ghost" circle starts approaching - the "ghost" circle disappears - a line indicates the movement direction). Bottom: Presentation of the stimulus in the level-2 saccades experiment (countdown at the start - the target shrinks - the target shrinks again - the target jumps).

Additionally, before the target started moving, a "ghost" circle appeared, indicating the start and speed of the movement. See Figure 7 for an example of the stimuli presentation in a "smooth" experiment.

In the "saccades" experiments, a line connected the current position of the target with its next position, i.e., the position to which the target jumped. To indicate the moment when the jump occurred, the circle got scaled down 3 times, completely disappearing the last time. The moment the circle disappeared was the moment the jump happened. This is shown in the lower part of Figure 7.

### B.1.7   Data Preprocessing

The suggested preprocessing pipeline consists of two steps, (i) imputation of missing values, and (ii) bandpass filtering. While the first step is computationally expensive, the second is efficient. To facilitate use of the data set, pre-processed data with imputed missing values is provided alongside the raw data and code to filter it.

### B.1.8   Missing Value Imputation

Missing values were recorded as zeros, which could also occur as valid values. Thus, the first step was to distinguish actual missing values from legitimate zero readings. To address this, zero values that occurred as part of an uninterrupted sequence of at least three zeros were identified as missing values. That means, a zero was flagged as missing if it was preceded, followed, or surrounded by two other zeros. This approach was adopted to ensure that isolated zero values were retained as valid data points, while sequences of zeros (which are unlikely to occur naturally in the data) were treated as missing data.

Once the missing values were identified, they were imputed using a Kalman smoother. In this case, a Seasonal Autoregressive Integrated Moving Average (SARIMA) model was used as the underlying state-space model for the Kalman smoother. The optimal SARIMA model for each recording was determined automatically using the `auto_arima` function from the `pmdarima` library. This function performs differencing tests to decide the order of differencing required for stationarity, together with a stepwise algorithm, as outlined by Hyndman & Khandakar (2008), to select the $p$, $q$, and seasonal $P$ and $Q$ orders based on the Akaike Information Criterion. The `auto_arima` function was applied with default parameters, with the exception of the seasonal lag, which was set to 5. This choice was made because SARIMA's seasonal component could explicitly model periodic noise patterns inherent in EEG recordings from our setup, such as the particularly strong 50 Hz power line signal, which, at a sampling rate of 256 Hz, corresponds to a period of approximately 5 samples ($256/50 = 5.12 \approx 5$). While the subsequently used frequency filters suppress noise at 50 Hz, our goal was to reconstruct the raw EEG recordings as accurately as possible. Simpler methods, such as linear interpolation, might yield similar results after application of the frequency filters. Figure 8 illustrates the effect of the missing value imputation on the data.

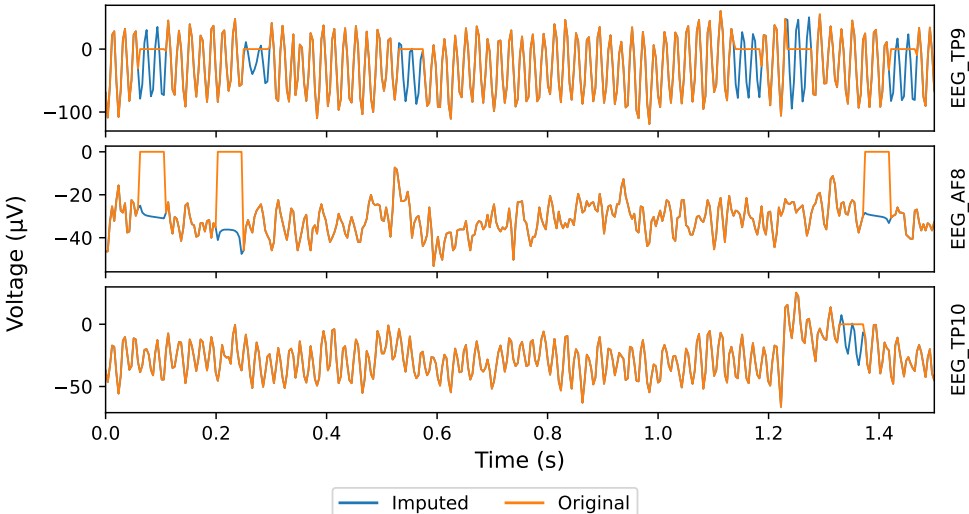

Figure 8: Exemplary imputation of missing values for recording "P045_01 level-1-saccades". Electrode AF7 did not contain missing values in this segment and is not displayed.

| Electrode | Imputed Values (%) | Windows With Imputation (%) |
|---|---|---|
| TP9 | 14.96 ($\pm$ 7.15) | 87.16 ($\pm$ 31.25) |
| AF7 | 0.22 ($\pm$ 4.38) | 0.32 ($\pm$ 4.95) |
| AF8 | 8.79 ($\pm$ 5.82) | 78.46 ($\pm$ 32.85) |
| TP10 | 2.52 ($\pm$ 4.63) | 36.72 ($\pm$ 22.04) |
| Overall | 6.62 ($\pm$ 8.03) | 50.66 ($\pm$ 43.01) |

Table 17: Mean ($\pm$ SD) percentage of imputed samples and windows with imputed values by electrode.

To quantitatively assess the effectiveness and impact of missing value imputation, we calculated several summary statistics across all electrodes and recordings. On average, the percentage of missing values per channel was 6.6% ($\pm$ 8.0%). The large standard deviation indicates that the distribution is skewed, with most recordings below average and few recordings with substantially more missing values. This effect is partially explained by differences between electrodes, see Table 17. EEG data were segmented into 1-second windows (256 samples), resulting in an average of 50.7% ($\pm$ 43.0%) of windows containing at least one missing value. Again, there were major differences between electrodes.

Direct comparison of signal variability between windows with and without imputed values is limited by substantial differences across recordings and by many recordings containing only one window type. As such, pooled or per-recording statistics may not accurately reflect the local impact of imputation.

### B.1.9 Filtering

As EEG data is highly prone to noise from various sources, filtering is a common preprocessing step in its analysis. Multiple frequency filters were applied subsequently. First, a 60 Hz notch filter was applied to remove the noise from the monitor refresh rate. This was followed by another 50 Hz notch filter to remove the noise from the power lines. Finally, a 0.5 Hz to 40 Hz Butterworth bandpass filter was applied to remove noise from other sources, such as muscle activity and, baseline drift. All filters were applied forwards and backwards to avoid phase distortion.

The filters were implemented using second-order sections (SOS) instead of directly applying the difference equation, as this was found to be more stable during testing. In Figure 9 the effect of applying the filters on the data is shown. It is evident that the vanilla filter causes the signal to diverge over time, while the

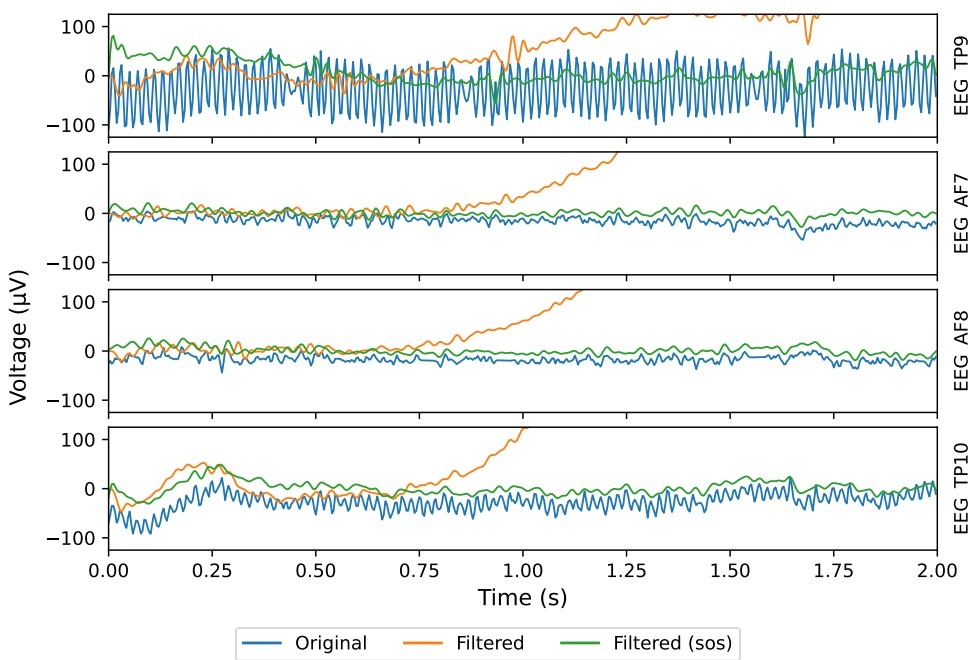

Figure 9: Frequency filters applied to the recording "P045_01 level-2-saccades", where missing values were imputed as described in the above section.

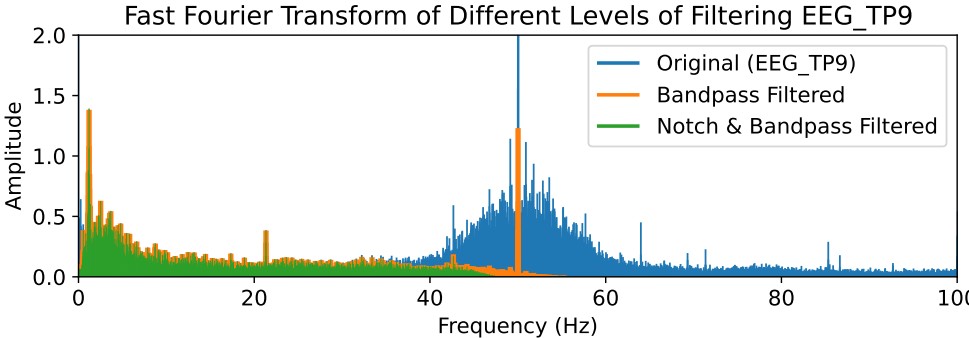

Figure 10: Bandpass filter applied to electrode TP9 of recording "P045_01 level-2-smooth".

SOS-filter keeps the signal stable. Both filters used the combination of 60 Hz and 50 Hz notch and 0.5 Hz to 40 Hz bandpass filter. However, the vanilla filter used an 8th order bandpass filter and applied all filters only forwards. The SOS-filter used a 4th order bandpass filter and applied all filters both forwards and backwards, doubling the order of the filter and correcting phase shift.

Even though the bandpass filter limits the data to the frequency range from 0.5 Hz to 40 Hz, which should exclude the 50 Hz and 60 Hz noise, we have found that applying the bandpass alone was insufficient to remove the noise entirely. This can be seen in Figure 10 where the noise from the power lines is still present in the data after applying the bandpass filter.

## B.2 Data Records

The "Consumer-Grade Electroencephalography and Eye-Tracking Dataset" is openly available on Zenodo (non-anonymous link)

In total, the dataset contains 11 hours and 45 minutes of EEG and eye tracking data. The total time recorded per paradigm and the duration in one session can be seen in Table 16.

The dataset is available as a single denormalized `CSV` file, which contains all the data collected during the sessions. The `CSV` file has a size of 1.42 GB and the following columns: `Participant_no`, `Task`, `Session_no`, `Timestamp`, `EEG_TP9`, `EEG_AF7`, `EEG_AF8`, `EEG_TP10`, `Gaze_x`, `Gaze_y`, `Stimulus_x`, `Stimulus_y`. Any samples recorded before or after the stimulus presentation are removed.

In addition to the `CSV` file, a "csv" and an "xdf" folder are provided. Inside both folders, there are four subfolders, one for every experimental paradigm. Furthermore, every subfolder contains two more folders called "train" and "test", which contain one file per recording. When machine learning models are created to predict the true eye position from the EEG data, only the training data should be used for fitting the model, while the test data should be used to evaluate the fitted model. The filename always follows the naming scheme PXXX_YY.csv or PXXX_YY.xdf, where XXX is the participant number and YY is the session number for this participant. Writing out the folder structure in a tree-like manner, one would get the following:

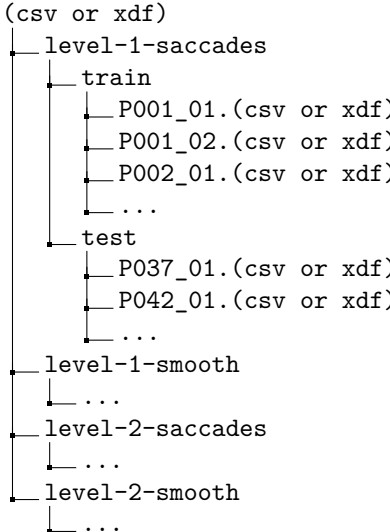

```
(csv or xdf)
└── level-1-saccades
    └── train
        └── P001_01.(csv or xdf)
        └── P001_02.(csv or xdf)
        └── P002_01.(csv or xdf)
        └── ...
    └── test
        └── P037_01.(csv or xdf)
        └── P042_01.(csv or xdf)
        └── ...
└── level-1-smooth
    └── ...
└── level-2-saccades
    └── ...
└── level-2-smooth
    └── ...
```

In the "csv" folder, the files are `CSV` files with columns `Timestamp`, `EEG_TP9`, `EEG_AF7`, `EEG_AF8`, `EEG_TP10`, `Gaze_x`, `Gaze_y`, `Stimulus_x`, `Stimulus_y`. A detailed description of the columns is provided in Table 18.

The "xdf" folder contains `XDF` files with the raw recorded data. Those files contain additional metadata of the multiple data streams. Users interested in more detailed, time-series structured data may find the `XDF` files useful.

The stimuli presentation software sends the current program status via a LSL stream. In particular, this includes a start and end marker. The first line of the `CSV` file is the first measurement after the start marker. Each line corresponds to a further EEG value. Since other signals (gaze position, program state, position of the target) do not change with the same frequency as the EEG data is recorded, these values always correspond to the most recent ones at this point in time. The last line of the `CSV` file is the last measurement before the end marker. In addition, the absolute positions in px from the `XDF` file are converted to positions relative to the center of the bounding box in mm.

Both the `CSV` file containing the complete dataset, and the "csv" and "xdf" folders are available on Zenodo.

### B.3 Technical Validation

In this section, we describe the technical validation of the data collected using a consumer-grade BCI headset and a webcam-based eye tracker. Given the nature of the experiments, there was a consistent trade-off between obtaining high-quality data and maintaining a realistic, non-laboratory setting. While the consumer-

| Source | Column Name | Data Type | Description |
|---|---|---|---|
| LSL | Timestamp | float | Timestamp from synchronized data streams |
| EEG | EEG_TP9 | float | EEG signal from electrode TP9 |
| | EEG_AF7 | float | EEG signal from electrode AF7 |
| | EEG_AF8 | float | EEG signal from electrode AF8 |
| | EEG_TP10 | float | EEG signal from electrode TP10 |
| Eye Tracker | Gaze_x | float | (estimated) $x$ coordinate of the gaze on screen |
| | Gaze_y | float | (estimated) $y$ coordinate of the gaze on screen |
| Presentation App | Stimulus_x | float | $x$ coordinate of the target on screen |
| | Stimulus_y | float | $y$ coordinate of the target on screen |

Table 18: Data structure description.

| Recording(s) | Reason for Exclusion |
|---|---|
| P002_01 | Webcam calibration quality declined significantly |
| P004_01 | `Gaze_x` attained unrealistically high values |
| P016_01 – P020_01 | A lot of missing values due to Bluetooth interference |
| P050_01 level-1-saccades | EEG disconnected before the recording finished |
| P062_01 – P067_01 | Very high AF8 readings due to faulty hardware |
| P079_01 | High TP9 and TP10 readings from wearing a headgear |

Table 19: List of recordings with known quality issues..

level hardware offered the advantage of a more natural and accessible data collection environment, it also posed challenges related to signal accuracy and reliability. This section outlines the steps taken to assess the quality of the data and address potential issues inherent in using such devices outside of controlled lab conditions.

### B.3.1 Known Quality Issues

During the data recording process, live EEG measurements and gaze data (tracked via the webcam) were continuously monitored. This real-time monitoring enabled early detection of recording issues, allowing the task to be restarted when no anomalies were observed. In certain cases, it was not possible to achieve reliable readings from specific electrodes. These instances, and similar issues, were documented during the recording sessions, and the corresponding recordings might be excluded from subsequent experiments.

In total, 14 out of 116 recorded sessions (12.1%) were flagged as having known quality issues (Table 19). The most common problems were faulty hardware leading to abnormal electrode readings (7.0%) and Bluetooth interference in EEG transmission affecting multiple consecutive sessions (4.3%). The affected recordings represent approximately 64 minutes, or about 9.3% of the total recording time. While some recordings may still contain usable segments, users should treat them with caution and consider excluding them or applying additional preprocessing depending on their analysis goals.

### B.3.2 EEG Data

The aim of the experiments was to create a representative dataset of EEG data recorded with consumer-grade hardware in a realistic environment. For the EEG data, we had to carefully decide when the signal-to-noise ratio became too low, even for this purpose. At the beginning of each recording, the signal quality was assessed visually. The EEG data was visualized using a viewer from the Python library `muselsl` Barachant et al. (2019), which applies a bandpass filter between 1 and 40 Hz by default. The filtered signal was then checked to ensure it was primarily below $30\,\mu\text{V}$. This naïve heuristic helped identify recordings whose signal-

to-noise ratio was too low even for a consumer-level BCI headset. A more systematic approach was avoided in order to collect representative and realistic EEG signals.

The data contains missing values due to incorrect transmission, which were stored as zeros. In the "Methods" section, we explain the approach that was used for the imputation of missing values.

### B.3.3 Stimulus Presentation

Several measures were implemented, to ensure that participants could effectively follow the target on the screen during the experiment. As described in the "Methods" section, the direction and timing of the target's movement could be easily predicted due to visual cues. The target was displayed as a circle that got scaled down three times before jumping to the next position. This gradual reduction in size should help the participants to better estimate the time of the next jump. Further visual cues allowed participants to anticipate the next location or direction of movement, such as a "ghost" dot and a line hinting in the latter direction, cf. Figure 7.

In addition to the visual cues, the participants tried out the various tasks in a trial run before the recording. This ensured that participants were familiar with the target's behavior and could follow it better.

These measures were crucial for maintaining the integrity of the data, while finding a balance between realistic conditions and the need for accurate eye tracking. By making the target visually clear, predictable, and easy to follow, we aimed to optimize the participant's ability to complete the task successfully, thus improving the reliability of the recorded data.

### B.3.4 Eye Tracking Data

The gaze position, as obtained from the eye tracking software, should not be mistaken for the "ground truth" of eye movements. In this study, the exact on-screen coordinates of the moving target were recorded with pixel-level precision. Given that participants were instructed and visually cued to follow this target closely throughout each trial, these target positions provide a practical proxy for true gaze location in subsequent analyses. Accordingly, when comparing EEG-based gaze predictions with webcam-based estimates, both should ideally be evaluated against the recorded target trajectory rather than directly against one another.

The data from the camera-based tracker was captured using a consumer-grade webcam and eye tracking software (GazePointer), which, although effective in non-laboratory settings, introduces certain inaccuracies and limitations. Its spatial accuracy depends on factors such as lighting conditions, camera angle, and calibration quality. External influences like head movement or changing illumination can further degrade performance. Published evaluations report an angular error between 1.4° and 1.9° Heck et al. (2023); Falch & Lohan (2024), which at our viewing distance of approximately 60 cm corresponds to about 15-20 mm on screen. By contrast, research-grade systems such as Tobii or EyeLink typically achieve <0.9° accuracy (< 9.5 mm at 60 cm) Dalrymple et al. (2018); Ehinger et al. (2019).

In addition to spatial imprecision, small temporal lags between actual eye movements and their detection can arise from both software processing delays and hardware limitations. While generally minor, these lags may introduce noticeable discrepancies during rapid saccades or other fast eye movements. These constraints are inherent to experiments based on consumer-grade hardware/software combinations, and must be considered when interpreting model performance using this dataset.

### B.3.5 Correlation Between Target and Gaze

The target positions correspond exactly to the location of the target on the screen, while the gaze positions are estimated by the eye tracking software. It is anticipated that the gaze will lag behind the target. To validate this relationship, we calculated the cross-correlation functions between the $x$ and $y$ coordinates of the gaze and target, respectively (see, e. g., Figure 11). Subsequently, we calculated the lag of maximal cross-correlation for each recording. The averages of these lags across all recordings are reported in Table 20.

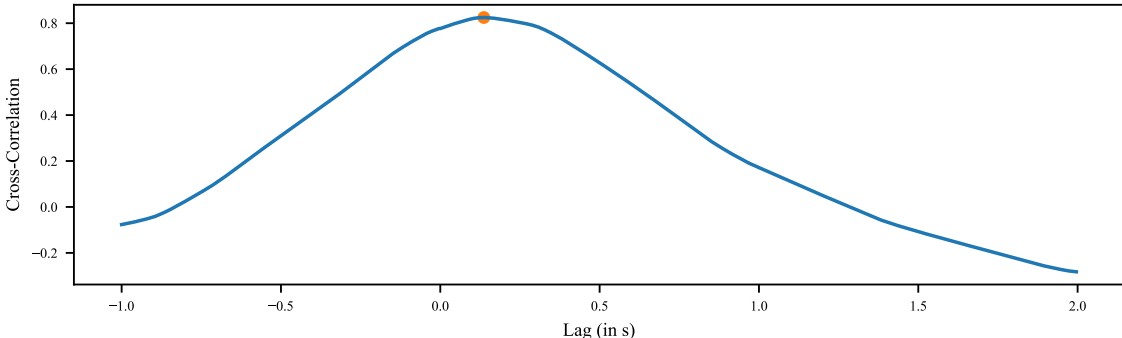

Figure 11: Cross-correlation function between $y$ coordinates of gaze and target for the level-1 "saccades" recording of participant 21.

| Experimental Paradigm | Avg. lag $(x)$ | Avg. lag $(y)$ |
|---|---|---|
| level-1-saccades | $0.18\,\text{s}$ ($\pm$ $0.13\,\text{s}$) | $0.23\,\text{s}$ ($\pm$ $0.13\,\text{s}$) |
| level-1-smooth | $0.36\,\text{s}$ ($\pm$ $0.20\,\text{s}$) | $0.38\,\text{s}$ ($\pm$ $0.20\,\text{s}$) |
| level-2-saccades | $0.16\,\text{s}$ ($\pm$ $0.20\,\text{s}$) | $0.19\,\text{s}$ ($\pm$ $0.18\,\text{s}$) |
| level-2-smooth | $0.26\,\text{s}$ ($\pm$ $0.16\,\text{s}$) | $0.32\,\text{s}$ ($\pm$ $0.18\,\text{s}$) |

Table 20: Average lag between target and gaze movement across all recordings, determined via cross-correlation function between stimulus and gaze coordinates per axis. Errors represent standard deviations across recordings' lag estimates within each paradigm.

### B.3.6 Extended Dependence Analysis

### B.4 Usage Notes

When working with the dataset, there are several aspects to take into account. EEG data is highly noisy even when recorded in a laboratory. The signals measured with consumer-grade hardware have a low signal-to-noise ratio. Therefore, it is recommended to preprocess the data first, or use the provided preprocessed data. Exemplary code to load the preprocessed data can be found in the GitHub repository (non-anonymous link).

To get started with the dataset, we recommend using `CSV` files. For users who are familiar with the `XDF` format, we recommend using the latter, as it contains additional information and metadata.

When evaluating EEG-based gaze predictions against webcam-based estimates, we recommend using the recorded target position as proxy for the ground truth, since webcam-derived gaze positions are subject to spatial inaccuracies (between 1.4° and 1.9°, corresponding to 15-20 mm at 60 cm).

When the dataset is used to train and evaluate machine learning models, only the training data should be used for model training, model selection and hyperparameter tuning. The test data should only be used for a final evaluation, and, in particular, not for model selection or hyperparameter tuning. For the latter tasks, the training data might be split into "training" and "validation" data.

It should be noted that, although demographic information such as gender and handedness was collected during the study, these attributes have been removed from the publicly released dataset to ensure participant anonymity. The participant pool is imbalanced (81% male; 88% right-handed), which could theoretically influence EEG-based eye tracking model performance if such factors affect ocular artifacts or gaze behavior. While users cannot directly control for these variables, the provided train-test split preserves the same underlying demographic ratios across both sets, helping to mitigate systematic bias between training and evaluation phases.

| Y \ w | 1 | 2 | 4 | 8 | 16 | 32 | 64 | 128 | 256 | 512 |
|---|---|---|---|---|---|---|---|---|---|---|
| *Panel A: level-1-saccades* | | | | | | | | | | |
| x-pos. Gaze | 0.050 (± 0.047) | 0.072 (± 0.061) | 0.120 (± 0.094) | 0.211 (± 0.157) | 0.398 (± 0.291) | 0.565 (± 0.371) | 0.667 (± 0.367) | 0.711 (± 0.372) | 0.858 (± 0.255) | 0.892 (± 0.232) |
| y-pos. Gaze | 0.049 (± 0.050) | 0.051 (± 0.055) | 0.074 (± 0.080) | 0.141 (± 0.146) | 0.317 (± 0.301) | 0.501 (± 0.398) | 0.619 (± 0.397) | 0.749 (± 0.345) | 0.862 (± 0.254) | 0.868 (± 0.276) |
| x-pos. Stimulus | 0.060 (± 0.049) | 0.077 (± 0.060) | 0.121 (± 0.093) | 0.208 (± 0.156) | 0.393 (± 0.289) | 0.564 (± 0.369) | 0.668 (± 0.365) | 0.713 (± 0.369) | 0.875 (± 0.255) | 0.888 (± 0.241) |
| y-pos. Stimulus | 0.051 (± 0.049) | 0.054 (± 0.056) | 0.079 (± 0.084) | 0.147 (± 0.149) | 0.322 (± 0.303) | 0.504 (± 0.400) | 0.620 (± 0.399) | 0.749 (± 0.344) | 0.888 (± 0.220) | 0.868 (± 0.272) |
| *Panel B: level-1-smooth* | | | | | | | | | | |
| x-pos. Gaze | 0.067 (± 0.063) | 0.094 (± 0.074) | 0.151 (± 0.108) | 0.252 (± 0.173) | 0.443 (± 0.291) | 0.611 (± 0.357) | 0.702 (± 0.348) | 0.779 (± 0.308) | 0.790 (± 0.321) | 0.925 (± 0.189) |
| y-pos. Gaze | 0.069 (± 0.058) | 0.075 (± 0.063) | 0.110 (± 0.093) | 0.189 (± 0.157) | 0.373 (± 0.294) | 0.552 (± 0.386) | 0.656 (± 0.384) | 0.743 (± 0.344) | 0.801 (± 0.318) | 0.913 (± 0.224) |
| x-pos. Stimulus | 0.067 (± 0.062) | 0.091 (± 0.073) | 0.144 (± 0.107) | 0.241 (± 0.171) | 0.431 (± 0.292) | 0.601 (± 0.363) | 0.691 (± 0.357) | 0.770 (± 0.315) | 0.776 (± 0.371) | 0.917 (± 0.219) |
| y-pos. Stimulus | 0.064 (± 0.057) | 0.070 (± 0.062) | 0.103 (± 0.092) | 0.179 (± 0.158) | 0.366 (± 0.296) | 0.547 (± 0.390) | 0.651 (± 0.387) | 0.741 (± 0.343) | 0.784 (± 0.430) | 0.919 (± 0.188) |
| *Panel C: level-2-saccades* | | | | | | | | | | |
| x-pos. Gaze | 0.074 (± 0.083) | 0.109 (± 0.111) | 0.170 (± 0.157) | 0.273 (± 0.225) | 0.441 (± 0.328) | 0.588 (± 0.367) | 0.692 (± 0.365) | 0.757 (± 0.367) | 0.837 (± 0.278) | 0.855 (± 0.393) |
| y-pos. Gaze | 0.080 (± 0.078) | 0.088 (± 0.093) | 0.124 (± 0.126) | 0.196 (± 0.192) | 0.364 (± 0.318) | 0.525 (± 0.390) | 0.637 (± 0.400) | 0.707 (± 0.414) | 0.851 (± 0.263) | 0.852 (± 0.343) |
| x-pos. Stimulus | 0.090 (± 0.073) | 0.101 (± 0.083) | 0.144 (± 0.112) | 0.225 (± 0.182) | 0.378 (± 0.308) | 0.524 (± 0.375) | 0.630 (± 0.379) | 0.721 (± 0.349) | 0.839 (± 0.288) | 0.863 (± 0.285) |
| y-pos. Stimulus | 0.088 (± 0.074) | 0.098 (± 0.082) | 0.136 (± 0.106) | 0.212 (± 0.174) | 0.367 (± 0.307) | 0.516 (± 0.380) | 0.619 (± 0.386) | 0.703 (± 0.366) | 0.861 (± 0.252) | 0.861 (± 0.279) |
| *Panel D: level-2-smooth* | | | | | | | | | | |
| x-pos. Gaze | 0.079 (± 0.068) | 0.109 (± 0.090) | 0.179 (± 0.132) | 0.214 (± 0.171) | 0.487 (± 0.300) | 0.653 (± 0.342) | 0.727 (± 0.339) | 0.801 (± 0.293) | 0.794 (± 0.326) | 0.938 (± 0.187) |
| y-pos. Gaze | 0.099 (± 0.075) | 0.104 (± 0.080) | 0.153 (± 0.112) | 0.229 (± 0.180) | 0.426 (± 0.313) | 0.594 (± 0.372) | 0.690 (± 0.367) | 0.779 (± 0.322) | 0.822 (± 0.301) | 0.920 (± 0.274) |
| x-pos. Stimulus | 0.076 (± 0.068) | 0.089 (± 0.077) | 0.134 (± 0.107) | 0.240 (± 0.180) | 0.417 (± 0.320) | 0.579 (± 0.384) | 0.678 (± 0.374) | 0.766 (± 0.331) | 0.797 (± 0.326) | 0.919 (± 0.212) |
| y-pos. Stimulus | 0.082 (± 0.068) | 0.087 (± 0.073) | 0.126 (± 0.099) | 0.290 (± 0.193) | 0.397 (± 0.318) | 0.559 (± 0.398) | 0.657 (± 0.395) | 0.751 (± 0.340) | 0.826 (± 0.295) | 0.917 (± 0.218) |

Table 21: Chatterjee's correlation coefficient (average and standard deviation over all recordings) between EEG windows of varying length $w$ and the $x$-/$y$-coordinate of target (stimulus or gaze).

A subset of the dataset (12.1% of all sessions, $\approx 9.3\%$ of total recording time) is flagged for potential quality concerns such as Bluetooth interference or faulty electrodes. These files are listed in Table 19. Depending on their tolerance for noise and missing data, users may wish to exclude these sessions entirely or apply additional preprocessing before model training.

Generally, we assume that methods from time series analysis are suitable for interpreting the data. The "smooth" experiments were specifically designed so that the dataset can serve as a benchmark in functional data analysis, and we expect that the corresponding methods are particularly suited for these recordings.

### B.5    Data Availability

The "Consumer-Grade Electroencephalography and Eye-Tracking Dataset" is openly available from Zenodo: Non-anonymous link (Zenodo record XXXXXXXX, DOI: xx.xxxx/zenodo.XXXXXXXX). The repository contains: (1) a single denormalized CSV file (1.42 GB) with all recordings and (2) separate "csv" and "xdf" folders organized by paradigm (level-1-saccades, level-1-smooth, level-2-saccades, level-2-smooth) with train/test subfolders and per-recording files (`PXXX_YY.csv` or `PXXX_YY.xdf`). The XDF files contain the raw multi-stream data and metadata. The CSV files provide time-aligned samples. The complete dataset comprises 11 hours 45 minutes of EEG and eye-tracking data.

### B.6    Code Availability

The code used for the recordings is written in Python and is publicly available on GitHub (Non-anonymous link).The code was developed with Python (3.11), Matplotlib (3.9.2), NumPy (1.26.4), Pandas (2.2.3), Pooch (1.8.2), SciPy (1.14.1), Statsmodels (0.14.4), and respective dependencies.

The repository contains the following key components:

- `stimuli-presentation-app`: Code for the application used to record the data.

- `analyse_data.py`: Script to calculate the lag between gaze and stimulus through the cross-correlation function, as described in the "Technical Validation" section.

- `impute_missing_values.py`: Script to impute missing values, as described in the "Methods" section.

- `xdf_to_csv.py`: Script to convert the raw data from XDF to CSV format.

Most importantly, the repository includes the `load_data` module, which provides functions to load and filter the data. The `load_data` functions leverage the `pooch` library in the backend to automatically fetch the required dataset from Zenodo, when it is called for the first time.

