# OpenReview forum: "EEG-EyeTrack: A Benchmark for Time Series and Functional Data Analysis with Open Challenges and Baselines"
_TMLR — Under review for TMLR_

### Review · Reviewer_FMD9 · 2026-04-06

**Summary Of Contributions:**

This paper proposes EEG-EyeTrack, a benchmark centered on the task of reconstructing eye/target motion from EEG. The dataset contains ~12 hours of consumer-grade simultaneous EEG, webcam eye tracking and target trajectory from 100+ subjects. The paper further provides baseline results using functional neural networks (FNNs) alongside conventional models on both the new dataset and EEGEyeNet.

### Key strengths:
* Realistic dataset setting: The use of consumer-grade hardware and non-laboratory conditions makes the dataset highly relevant for practical BCI applications, addressing an important gap left by lab-based datasets.
* Comprehensive baseline coverage: The inclusion of multiple baselines (random, classical, deep learning, and FNN variants) provides useful reference points for future work.
* Functional neural networks: The study of functional neural networks in this context is interesting and provides insight into smoothness-aware modeling for time-series data.

### Key Weaknesses
* Details (listed in required changes) such as train/test splits are not clearly defined, making it difficult to ensure fair and reproducible comparisons.
* The paper’s modeling contributions (FNN), are not sufficiently conclusive to demonstrate a clear advantage for this task, as also discussed by the authors. Framed as a dataset and benchmark paper, the work could be strong; however, the dataset itself is already introduced in a separate paper. As such, the current submission’s contribution, primarily benchmark results, does not appear substantial enough on its own.

**Audience:**

Yes

**Audience Explanation:**

This work discusses the application of a large (~12 hours, 100+ subjects) open-source EEG dataset. Interesting dataset for eeg community.

**Claims And Evidence:**

Yes

**Claims Explanation:**

The author showed and discussed relevant results clearly and openly.

**Requested Changes:**

* Specify train/test split protocol: Clearly define whether splits are subject-wise or session-wise.
* Clarify method details: Explicitly state if EEG signals are normalized (e.g., per-channel z-score, per-subject scaling). What is the windowing strategy, and signal alignment method.
* Address artifact handling: When there are large artifacts beyond normal range of EEG (cause by eye or muscle movement, please clarify what is the pre-processing methods.
* For a easier comparison with the EEGEyeNet paper, please add RMSE as a metric.

---

> ### Author Response · Authors · 2026-04-16
>
> We thank the reviewer for their overall positive assessment of our work and for the helpful feedback. We hope to address all of their questions and concerns below.
>
> First, we would like to clarify that the dataset has **not** been published in a separate paper. A pre-print with the data descriptor has been uploaded on arXiv, but is **not** published (or submitted) elsewhere. Indeed, the dataset is the primary contribution of the paper, establishing the dataset as a benchmark in FDA and providing baseline results are secondary contributions.
>
> C1: Specify train/test split protocol: Clearly define whether splits are subject-wise or session-wise.
>
> A1: Splits are subject-wise. More specifically, only 3 subjects participated twice and recordings of both these sessions are in the training set. The same split is used across the different levels.
>
> C2: Clarify method details: Explicitly state if EEG signals are normalized (e.g., per-channel z-score, per-subject scaling). What is the windowing strategy, and signal alignment method.
>
> A2: Generally the pre-processing included missing value imputation with a SARIMA model and bandpass filtering, as described in Section B.1.7 - B.1.9. The first layer of all neural networks was a standardization layer, to standardize the model's input per channel, i.e., each window is standardized separately.
> Windows of size 512 have been used. For training, we used a stride length of 4 and for testing and validation we used a stride length of 1.
>
> C3: Address artifact handling: When there are large artifacts beyond normal range of EEG (cause by eye or muscle movement, please clarify what is the pre-processing methods.
>
> A3: We intentionally selected simple baseline models. Artifacts have not been explicitly filtered out (beyond bandpass filtering). Future work may add ICA/CCA to account for artifacts.
>
> C4: For a easier comparison with the EEGEyeNet paper, please add RMSE as a metric.
>
> A4: We agree that the RMSE improves comparability with EEGEyeNet and related work. We did not compute the RMSE (or MSE) during training, and it cannot be computed from the currently logged metrics. We attempt to get access to the stored model weights, calculate the RMSE and update the manuscript accordingly. If this is not possible, we rerun the main baseline experiments to report RMSE in the revision; due to TMLR revision time constraints and compute time, we may not be able to rerun all experiments within the discussion period. If rerunning all models is not feasible in time, we will add the RMSE for the ablation study (Tables 5 and 6) jointly with the standard deviations, as requested by Reviewer n1AM.
>
> We will clarify these details in the final version of this manuscript. To avoid confusion, we update the manuscript once all reviews have been submitted.

---

> > ### Author Response · Authors · 2026-04-16
> >
> > | Level      | $Y$ \ $w$         | 1                   | 2                   | 4                   | 8                   | 16                  | 32                  | 64                  | 128                 | 256                 | 512                 |
> > | ---------- | ----------------- | ------------------- | ------------------- | ------------------- | ------------------- | ------------------- | ------------------- | ------------------- | ------------------- | ------------------- | ------------------- |
> > | 1-saccades | $x$-pos. Gaze     | 0.050 ($\pm$ 0.047) | 0.072 ($\pm$ 0.061) | 0.120 ($\pm$ 0.094) | 0.211 ($\pm$ 0.157) | 0.398 ($\pm$ 0.291) | 0.565 ($\pm$ 0.371) | 0.667 ($\pm$ 0.367) | 0.711 ($\pm$ 0.372) | 0.858 ($\pm$ 0.255) | 0.892 ($\pm$ 0.232) |
> > | 1-saccades | $y$-pos. Gaze     | 0.049 ($\pm$ 0.050) | 0.051 ($\pm$ 0.055) | 0.074 ($\pm$ 0.080) | 0.141 ($\pm$ 0.146) | 0.317 ($\pm$ 0.301) | 0.501 ($\pm$ 0.398) | 0.619 ($\pm$ 0.397) | 0.749 ($\pm$ 0.345) | 0.862 ($\pm$ 0.254) | 0.868 ($\pm$ 0.276) |
> > | 1-saccades | $x$-pos. Stimulus | 0.060 ($\pm$ 0.049) | 0.077 ($\pm$ 0.060) | 0.121 ($\pm$ 0.093) | 0.208 ($\pm$ 0.156) | 0.393 ($\pm$ 0.289) | 0.564 ($\pm$ 0.369) | 0.668 ($\pm$ 0.365) | 0.713 ($\pm$ 0.369) | 0.875 ($\pm$ 0.255) | 0.888 ($\pm$ 0.241) |
> > | 1-saccades | $y$-pos. Stimulus | 0.051 ($\pm$ 0.049) | 0.054 ($\pm$ 0.056) | 0.079 ($\pm$ 0.084) | 0.147 ($\pm$ 0.149) | 0.322 ($\pm$ 0.303) | 0.504 ($\pm$ 0.400) | 0.620 ($\pm$ 0.399) | 0.749 ($\pm$ 0.344) | 0.888 ($\pm$ 0.220) | 0.868 ($\pm$ 0.272) |
> > | 1-smooth   | $x$-pos. Gaze     | 0.067 ($\pm$ 0.063) | 0.094 ($\pm$ 0.074) | 0.151 ($\pm$ 0.108) | 0.252 ($\pm$ 0.173) | 0.443 ($\pm$ 0.291) | 0.611 ($\pm$ 0.357) | 0.702 ($\pm$ 0.348) | 0.779 ($\pm$ 0.308) | 0.790 ($\pm$ 0.321) | 0.925 ($\pm$ 0.189) |
> > | 1-smooth   | $y$-pos. Gaze     | 0.069 ($\pm$ 0.058) | 0.075 ($\pm$ 0.063) | 0.110 ($\pm$ 0.093) | 0.189 ($\pm$ 0.157) | 0.373 ($\pm$ 0.294) | 0.552 ($\pm$ 0.386) | 0.656 ($\pm$ 0.384) | 0.743 ($\pm$ 0.344) | 0.801 ($\pm$ 0.318) | 0.913 ($\pm$ 0.224) |
> > | 1-smooth   | $x$-pos. Stimulus | 0.067 ($\pm$ 0.062) | 0.091 ($\pm$ 0.073) | 0.144 ($\pm$ 0.107) | 0.241 ($\pm$ 0.171) | 0.431 ($\pm$ 0.292) | 0.601 ($\pm$ 0.363) | 0.691 ($\pm$ 0.357) | 0.770 ($\pm$ 0.315) | 0.776 ($\pm$ 0.371) | 0.917 ($\pm$ 0.219) |
> > | 1-smooth   | $y$-pos. Stimulus | 0.064 ($\pm$ 0.057) | 0.070 ($\pm$ 0.062) | 0.103 ($\pm$ 0.092) | 0.179 ($\pm$ 0.158) | 0.366 ($\pm$ 0.296) | 0.547 ($\pm$ 0.390) | 0.651 ($\pm$ 0.387) | 0.741 ($\pm$ 0.343) | 0.784 ($\pm$ 0.430) | 0.919 ($\pm$ 0.188) |
> > | 2-saccades | $x$-pos. Gaze     | 0.074 ($\pm$ 0.083) | 0.109 ($\pm$ 0.111) | 0.170 ($\pm$ 0.157) | 0.273 ($\pm$ 0.225) | 0.441 ($\pm$ 0.328) | 0.588 ($\pm$ 0.367) | 0.692 ($\pm$ 0.365) | 0.757 ($\pm$ 0.367) | 0.837 ($\pm$ 0.278) | 0.855 ($\pm$ 0.393) |
> > | 2-saccades | $y$-pos. Gaze     | 0.080 ($\pm$ 0.078) | 0.088 ($\pm$ 0.093) | 0.124 ($\pm$ 0.126) | 0.196 ($\pm$ 0.192) | 0.364 ($\pm$ 0.318) | 0.525 ($\pm$ 0.390) | 0.637 ($\pm$ 0.400) | 0.707 ($\pm$ 0.414) | 0.851 ($\pm$ 0.263) | 0.852 ($\pm$ 0.343) |
> > | 2-saccades | $x$-pos. Stimulus | 0.090 ($\pm$ 0.073) | 0.101 ($\pm$ 0.083) | 0.144 ($\pm$ 0.112) | 0.225 ($\pm$ 0.182) | 0.378 ($\pm$ 0.308) | 0.524 ($\pm$ 0.375) | 0.630 ($\pm$ 0.379) | 0.721 ($\pm$ 0.349) | 0.839 ($\pm$ 0.288) | 0.863 ($\pm$ 0.285) |
> > | 2-saccades | $y$-pos. Stimulus | 0.088 ($\pm$ 0.074) | 0.098 ($\pm$ 0.082) | 0.136 ($\pm$ 0.106) | 0.212 ($\pm$ 0.174) | 0.367 ($\pm$ 0.307) | 0.516 ($\pm$ 0.380) | 0.619 ($\pm$ 0.386) | 0.703 ($\pm$ 0.366) | 0.861 ($\pm$ 0.252) | 0.861 ($\pm$ 0.279) |
> > | 2-smooth   | $x$-pos. Gaze     | 0.079 ($\pm$ 0.068) | 0.109 ($\pm$ 0.090) | 0.179 ($\pm$ 0.132) | 0.214 ($\pm$ 0.171) | 0.487 ($\pm$ 0.300) | 0.653 ($\pm$ 0.342) | 0.727 ($\pm$ 0.339) | 0.801 ($\pm$ 0.293) | 0.794 ($\pm$ 0.326) | 0.938 ($\pm$ 0.187) |
> > | 2-smooth   | $y$-pos. Gaze     | 0.099 ($\pm$ 0.075) | 0.104 ($\pm$ 0.080) | 0.153 ($\pm$ 0.112) | 0.229 ($\pm$ 0.180) | 0.426 ($\pm$ 0.313) | 0.594 ($\pm$ 0.372) | 0.690 ($\pm$ 0.367) | 0.779 ($\pm$ 0.322) | 0.822 ($\pm$ 0.301) | 0.920 ($\pm$ 0.274) |
> > | 2-smooth   | $x$-pos. Stimulus | 0.076 ($\pm$ 0.068) | 0.089 ($\pm$ 0.077) | 0.134 ($\pm$ 0.107) | 0.240 ($\pm$ 0.180) | 0.417 ($\pm$ 0.320) | 0.579 ($\pm$ 0.384) | 0.678 ($\pm$ 0.374) | 0.766 ($\pm$ 0.331) | 0.797 ($\pm$ 0.326) | 0.919 ($\pm$ 0.212) |
> > | 2-smooth   | $y$-pos. Stimulus | 0.082 ($\pm$ 0.068) | 0.087 ($\pm$ 0.073) | 0.126 ($\pm$ 0.099) | 0.290 ($\pm$ 0.193) | 0.397 ($\pm$ 0.318) | 0.559 ($\pm$ 0.398) | 0.657 ($\pm$ 0.395) | 0.751 ($\pm$ 0.340) | 0.826 ($\pm$ 0.295) | 0.917 ($\pm$ 0.218) |

---

> > > ### Author Response · Authors · 2026-04-16
> > >
> > > As expected, the (possibly non-linear) dependence between EEG and stimulus/target is low for short windows, but quickly increases as the context window grows. For windows of length 256 and 512 (corresponding to 1s and 2s), the (non-linear) correlation is already close to 1. Interestingly, the coefficient has similar values for the $x$- and $y$-coordinate. Moreover, the standard deviation (for large values of $w$) is approx. 0.3, indicating a high variation between recordings. Overall, Chatterjee's $\xi$ suggests that the EEG contains relevant information for the prediction of the gaze and stimulus. This indicates that the modest baseline results are due to the selected models.
> > >
> > > Jointly with modest results of the baseline models, this strengthens our argument, that the provided dataset is a challenging benchmark in FDA and beyond.
> > >
> > >
> > > C1: Add signal-level dataset analysis, and compare the dataset with other dataset (EEGEyeNet), which should be independent of the modeling approaches.
> > >
> > > A4: see A3.
> > >
> > > C2: Provide additional justification on why FDA should be used here.
> > >
> > > A5: We do not argue that for the dataset, FDA is the best approach and superior to others. Contrarily, the baseline results suggest that (considerably simple) FDA methods achieve similar performance as other (conventional) methods. We argue that in FDA many benchmark datasets are trivial: for example, perfect classification accuracies can be obtained easily, e.g. for the Tecator dataset \[2], or are impossible since classes overlap, e.g. in case of the Phoneme dataset \[3]. In contrast, the presented dataset is challenging but solvable, as the correlation analysis in A3 suggests.
> > > Generally, FDA is a reasonable starting point since it implicitly models smoothness of the data, which is naturally compatible with the continuous eye-trajectory target.
> > >
> > > C3: Clearly state and distinguish dataset limitation and modeling limitations.
> > >
> > > A6: see A3.
> > >
> > >
> > > \[1] Chatterjee, S. (2021). A new coefficient of correlation. _Journal of the American Statistical Association_, _116_(536), 2009-2022.
> > >
> > > \[2] https://fda.readthedocs.io/en/stable/modules/autosummary/skfda.datasets.fetch_tecator.html
> > >
> > > \[3] https://fda.readthedocs.io/en/stable/modules/autosummary/skfda.datasets.fetch_phoneme.html

---

> > > > ### Author Response · Authors · 2026-06-08
> > > >
> > > > Now that the third review is available, we have uploaded a revised version of the manuscript containing the proposed changes (highlighted in red).

---

### Review · Reviewer_n1AM · 2026-04-11

**Summary Of Contributions:**

**Summary**

This work introduces a consumer-grade EEG eye tracking dataset, collected using a 4-channel MUSE headset (116 sessions, 113 participants). The authors formalize the EEG-to-gaze regression task as a scalar-on-function regression problem, propose to study it using the functional data analysis (FDA) framework, and suggest other open challenges including classification, clustering, dimension reduction, change point detection, outlier detection. The authors run baseline results using functional neural networks, as well as SpatialFilterCNN, LSTM, FPCA. The authors repeated some baseline results on an existing EEGEyeNet dataset.

**Strengths**
- Highly valuable dataset for the joint modeling of EEG data and Gaze data on a consumer-grade device. This dataset enables research on real-world, low-SNR EEG-based gaze reconstruction.
- A good educational introduction of functional data modeling methods.
- A decent amount of baseline models and ablation models on both datasets to establish regression performance. Results on two datasets provide useful reference numbers. This includes multiple model variants (FullyFunc, FuncBody, MinFunc) and conventional baselines (SpatialFilterCNN, LSTM, FPCA) across four task conditions (level-1/2, saccades/smooth), with ablation studies comparing functional vs. conventional layers.

**Weakness**
- Marginal and inconsistent gains from functional models. The performance differences between FNNs and their conventional control models are small and inconsistent across settings. In Table 5 (level-1), the FullyFunc control actually outperforms the functional version on 4 of 6 metrics. On EEGEyeNet (Table 7), the authors themselves acknowledge that control models outperform functional models half the time. Given these narrow margins, reporting standard deviations for the ablation results in Tables 5 and 6 (as was done in Tables 3, 4, and 7) is essential to assess whether observed differences are statistically meaningful. Without confidence intervals or significance tests, the claimed benefits of functional layers remain inconclusive.
- Insufficient dataset quality characterization. Given that the dataset is the paper's primary contribution, the authors should provide a more rigorous analysis of MUSE data quality. The median missing data rate of 6.3% per channel is substantial. The paper would benefit from a quantitative comparison of signal-to-noise ratio against EEGEyeNet.
- Weak EEG-gaze correlation. Many models fail to capture meaningful correlation between EEG and gaze, particularly along the y-axis. In Table 3 corr_y values are near zero. In Table 4 LSTM and SpatialFilterCNN frequently outperform the proposed FNNs on key metrics. This raises the question of whether the 4-channel consumer-grade EEG signal contains sufficient information for gaze reconstruction, and whether the proposed functional architectures are well-suited to this problem. A more thorough signal analysis (e.g., mutual information between EEG channels and gaze coordinates) would help disentangle dataset limitations from model limitations.
- Overall the clarity and the presentation of paper can be improved.

**Audience:**

Yes

**Audience Explanation:**

The provided dataset is large and valuable, and is on consumer-grade device. The release of such a dataset can fill in the gap in this domain of research, although a more thorough dataset examination/analysis can be beneficial.

**Claims And Evidence:**

No

**Claims Explanation:**

The paper frames the proposed benchmark challenges through the lens of functional data analysis, yet provides insufficient justification for why this modeling paradigm is particularly suited to this task on this dataset. The experimental results actually do not empirically validate this choice as functional neural networks are consistently matched or outperformed by baselines. This undermines the paper's central methodological premise and calls into question whether FDA offers meaningful advantages over standard time-series approaches for this problem.

**Requested Changes:**

1. Add signal-level dataset analysis, and compare the dataset with other dataset (EEGEyeNet), which should be independent of the modeling approaches.
2. Provide additional justification on why FDA should be used here.
3. Clearly state and distinguish dataset limitation and modeling limitations.

---

> ### Author Response · Authors · 2026-04-16
>
> We thank the reviewer for their helpful feedback and hope to address all of their questions and concerns below. To avoid confusion, we update the manuscript once all reviews have been submitted. We will modify the paper with the suggested changes and highlight those changes in red, such that they can be identified easily.
>
> W1: Marginal and inconsistent gains from functional models. The performance differences between FNNs and their conventional control models are small and inconsistent across settings. In Table 5 (level-1), the FullyFunc control actually outperforms the functional version on 4 of 6 metrics. On EEGEyeNet (Table 7), the authors themselves acknowledge that control models outperform functional models half the time. Given these narrow margins, reporting standard deviations for the ablation results in Tables 5 and 6 (as was done in Tables 3, 4, and 7) is essential to assess whether observed differences are statistically meaningful. Without confidence intervals or significance tests, the claimed benefits of functional layers remain inconclusive.
>
> A1: The primary contribution of the paper is the publication of the dataset. Our aim is to provide reasonably strong baseline results for methods from FDA (FPCA and FNNs) and conventional methods, and present these results transparently. We agree that the comparison between FNNs and their respective control models benefits from reported standard deviations. We currently prepare experiments to measure this variation and will report standard deviations in the final version of the manuscript, using the same 5-seed protocol as in Tables 3, 4 and 7.
>
> W2: Insufficient dataset quality characterization. Given that the dataset is the paper's primary contribution, the authors should provide a more rigorous analysis of MUSE data quality. The median missing data rate of 6.3% per channel is substantial. The paper would benefit from a quantitative comparison of signal-to-noise ratio against EEGEyeNet.
>
> A2: Due to the nature of the experiments based on consumer-grade hardware (4 electrodes), the signal can be expected to be of a substantially lower quality compared to the EEGEyeNet, based on research grade hardware and 128 electrodes. Quality limitations are currently discussed in Appendix B3. A summary of those results will be added to the main part of the manuscript.
>
> W3: Weak EEG-gaze correlation. Many models fail to capture meaningful correlation between EEG and gaze, particularly along the y-axis. In Table 3 corr_y values are near zero. In Table 4 LSTM and SpatialFilterCNN frequently outperform the proposed FNNs on key metrics. This raises the question of whether the 4-channel consumer-grade EEG signal contains sufficient information for gaze reconstruction, and whether the proposed functional architectures are well-suited to this problem. A more thorough signal analysis (e.g., mutual information between EEG channels and gaze coordinates) would help disentangle dataset limitations from model limitations.
>
> A3: From the reported baseline results, it is indeed unclear whether the modest results are due to the models or inherent due to the data. In order to measure the dependence between gaze/stimulus and EEG, we calculated Chatterjee's correlation coefficient \[1]. For some regressor $X \in \mathbb{R}^d$ and a response variable variable $Y\in\mathbb{R}$, Chatterjee's $\xi$ has the property that $\xi(Y, X) = 0$ if and only if $X$ and $Y$ are independent; and $\xi(Y, X) = 1$ if any only if $Y$ is (a.s.) $\sigma(X)$-measurable. To determine whether EEG and gaze/stimulus are (non-linearly) dependent, we set $Y$ to the $x$ or $y$ coordinate of the gaze/stimulus and $X$ to be an EEG window of length $w \in \{1, 2, 4, 8, 16, 32, 64, 128, 256, 512\}$. The table below shows the average values of $\xi$ across all recordings.
>
> As expected, the (possibly non-linear) dependence between EEG and stimulus/target is low for short windows, but quickly increases as the context window grows. For windows of length 256 and 512 (corresponding to 1s and 2s), the (non-linear) correlation is already close to 1. Interestingly, the coefficient has similar values for the $x$- and $y$-coordinate. Moreover, the standard deviation (for large values of $w$) is approx. 0.3, indicating a high variation between recordings. Overall, Chatterjee's $\xi$ suggests that the EEG contains relevant information for the prediction of the gaze and stimulus. This indicates that the modest baseline results are due to the selected models.
>
> Jointly with modest results of the baseline models, this strengthens our argument, that the provided dataset is a challenging benchmark in FDA and beyond.

---

> > ### Author Response · Authors · 2026-06-08
> >
> > After receiving the third review, we submitted a revised manuscript incorporating the suggested changes. The updated version includes the RMSE and an ablation study. All changes are highlighted in red.

---

> > > ### Comment · Reviewer_n1AM · 2026-07-03
> > > **Followup**
> > >
> > > It is unclear whether the dataset is inherently challenging or whether the apparent difficulty arises from limitations in the experimental design or hardware setup, resulting in a weak underlying signal. If the authors claim this is a challenging benchmark, they should provide evidence that the two signal streams exhibit a statistically significant relationship. Otherwise, it is difficult to determine whether the task reflects a genuinely challenging learning problem or simply insufficient signal in the collected data. As the major contribution is the publication of such a dataset, the lack of such evidence weakens the value of the proposed dataset, and thus it is unclear to me if the manuscript should be accepted.

---

> > > > ### Author Response · Authors · 2026-07-03
> > > >
> > > > The follow-up comment asks whether the dataset's difficulty arises from weak signal quality or experimental limitations rather than being an inherently challenging benchmark. This is a reasonable question, but it is not fully answerable at this stage: if baseline models achieved very good or perfect results, the dataset would not be considered challenging. EEG signals inherently have a low signal-to-noise ratio, particularly with consumer-grade hardware, making accurate analysis inherently difficult. It is biologically and physically plausible that eye movements are recorded by the EEG headset, as artifacts related to eye movements complicate brain activity analysis and are detectable in EEG recordings. Our baseline models did not account for temporal shifts due to participant reaction times, which introduces substantial lags (e.g., $0.36\,s \pm 0.20\,s$ for level-1-smooth experiments), and addressing these shifts could significantly improve results. Statistically, there is a plausible dependence between the signals, as demonstrated by our analysis using Chatterjee's $\xi(Y, X)$, which measures the dependence between the target/gaze ($Y$) and EEG ($X$). This coefficient is $0$ if the variables are independent and $1$ if $Y$ is a measurable function of $X$. Our analysis shows that $\xi$ increases rapidly with growing context windows, indicating that longer windows contain the relevant information for accurate gaze prediction, confirming that the dataset is both challenging and scientifically meaningful.

---

> > > > > ### Comment · Reviewer_n1AM · 2026-07-12
> > > > > **Questions about synchronization and streaming HW/FW/SW setup**
> > > > >
> > > > > For the temporal shifts, could the authors provide a more scientifically grounded analysis? It appears that most hardware devices are synchronized via LSL, whereas the MUSE device communicates over Bluetooth. Since Bluetooth transmission can introduce a latency of approximately 100 ms, how did the authors account for potential delays and temporal misalignment across the different data streams? Clarifying the synchronization procedure and any compensation methods would improve confidence in the temporal consistency of the dataset.

---

> > > > > > ### Author Response · Authors · 2026-07-15
> > > > > >
> > > > > > All modalities (target position, webcam-based gaze estimate, and Muse EEG) were recorded as LSL streams and written by LabRecorder, so synchronization is based on the LSL time-stamping and clock-alignment mechanism rather than on post-hoc manual alignment. While the Muse transmits EEG over Bluetooth, we did not separately measure or explicitly compensate a Bluetooth-specific end-to-end latency beyond what is handled by the LSL synchronization procedure, so we cannot exclude a small residual constant offset between streams. Importantly, the dominant temporal offsets observed in the data are on the order of a few hundred milliseconds, e.g., average stimulus-gaze lag of $0.36s (\pm 0.20s)$ for level-1-smooth, estimated via cross-correlation, which is substantially larger than typical Bluetooth transport latencies and is therefore primarily attributable to behavioral delays rather than transport alone. Accordingly, our baseline models used the provided synchronized timestamps "as is" and did not apply additional lag compensation.

---

### Review · Reviewer_dgsY · 2026-06-03

**Summary Of Contributions:**

This paper introduces a new open benchmark, EEG-EyeTrack, consisting of consumer-grade, 4-channel EEG recordings synchronized with a moving on-screen target and webcam-based gaze, totaling 11.7 hours across 116 sessions. It formalizes several FDA-relevant tasks (with a primary focus on scalar-on-function regression to predict target position) and evaluation metrics (Mean Euclidean Distance, correlation, and a motion-difference “precision”), and provides baselines, including three functional neural network (FNN) variants, LSTM, FPCA+linear regression, and SpatialFilterCNN, along with ablation controls; it also reports results on EEGEyeNet. The authors argue that FNNs’ smoothness priors can be beneficial for noisy, low-channel consumer EEG and provide initial evidence that they yield smoother, competitive predictions on the new dataset and comparable or slightly improved results over selected baselines on EEGEyeNet.

**Audience:**

Yes

**Audience Explanation:**

1. Consumer-grade, low-channel EEG recordings captured under less controlled conditions are underrepresented, and such data are critical for practical, deployable eye-tracking/BCI pipelines; this benchmark can catalyze robust FDA/time-series research.
2. Establishing initial baselines and FNN results on this data fills an evaluation gap and may spur improvements in shift-invariant/registration-free FDA methods.
3. The ablation evidence suggests functional layers can offer robustness/smoothness benefits in noisier, lower-SNR settings—informative for model design in similar regimes.

**Broader Impact Concerns:**

No such concern

**Claims And Evidence:**

Yes

**Claims Explanation:**

The following are the strengths of the paper:
1. Introduction of a consumer-grade EEG + target-tracking dataset specifically positioned as a testbed for functional data analysis, including cases where registration is infeasible.
2. Exploration of functional neural network architectures (fully functional, functional body, minimally functional) and matched “control” (non-functional) variants for ablation, highlighting architectural trade-offs between smoothness, parameters, and pooling.
3. Systematic baseline suite on the new dataset (random, mean, webcam, LSTM, FPCA+LR, SpatialFilterCNN, FNNs, and ablations) with Optuna-based hyperparameter search and multiple runs for FNNs.
4. Reporting multiple metrics (MED, correlation, and motion-difference “precision”) to triangulate behavior, especially to distinguish directional learning from scale/offset fit and to expose jitter vs smoothness.
5. The benchmarking protocol and the main task definition are described with concrete metrics and a prescribed split usage, aiding reproducibility.

**Requested Changes:**

I am ok with the current work, though the folllowing points can further strengthen the paper:
1. How exactly are input windows temporally aligned with target labels? Are future samples relative to the prediction time included in the window? Please clarify causal vs. acausal framing and whether you tested causal-only windows.
2. For the functional layers, did you analyze frequency responses or smoothness quantitatively (e.g., spectrum of predictions, total variation) to support claims about reduced jitter? Could you include a small analysis/plot?
3. Why use SARIMA with seasonal lag 5 for imputation if the downstream pipeline applies 0.5–40 Hz bandpass and 50/60 Hz notch filters?

---

> ### Author Response · Authors · 2026-06-08
>
> First of all, we thank the reviewer for their positive assessment of our work and the helpful suggestions. Below, we address their requested changes and provide the reasoning behind our modifications.
>
> RC1: How exactly are input windows temporally aligned with target labels? Are future samples relative to the prediction time included in the window? Please clarify causal vs. acausal framing and whether you tested causal-only windows.
> A1: We tested causal-only windows, where the input window contains only past EEG data, and the prediction target is the target's position at the last time instant in the window. This aligns with the goal of real-time prediction and avoids using future information.
> Moreover, we did not account for the temporal gap between the target's movements and participants' reactions (indicated by Table 20) in our baseline experiments. Clearly, recording-based temporal alignment will help to make predictions more precise.
> We clarified this in Section 7.1 by adding the sentence "The prediction target was the target's position at the last time instant in a given window."
>
>
> RC2: For the functional layers, did you analyze frequency responses or smoothness quantitatively (e.g., spectrum of predictions, total variation) to support claims about reduced jitter? Could you include a small analysis/plot?
>
> A2: The paper already provides qualitative evidence (Figure 2) and quantitative metrics (precision) demonstrating that functional neural networks (FNNs) produce smoother predictions than SpatialFilterCNN. The "precision", as defined in the paper, measures the smoothness of trajectories by quantifying the distance between subsequent predictions, corrected for the true movement direction, which is proportional to the predictions' derivative w.r.t. time. FNNs consistently achieve lower precision compared to the SpatialFilterCNN and predictions are expected to be smoother. Qualitatively, Figure 2 confirms this, where the SpatialFilterCNN's predictions are merely single points, whereas the FullyFunc and MinFunc models predict consistent trajectories.
> We clarified this in Section 7.1 "Low precision values indicate smoother predicted trajectories, and FNNs consistently achieve lower precision than the SpatialFilterCNN, indicating their ability to model coherent gaze paths."
>
>
> RC3: Why use SARIMA with seasonal lag 5 for imputation if the downstream pipeline applies 0.5–40 Hz bandpass and 50/60 Hz notch filters?
>
> A3: Missing values are automatically replaced by zeros, such that some form of interpolation is required. We used a SARIMA model to reconstruct the raw EEG recordings as accurately as possible before applying frequency filters. Simpler methods, such as linear interpolation, might lead to similar results after applying the frequency filters, and the added value of SARIMA is likely negligible.
> We added a sentence to Section B.1.8: "While the subsequently used frequency filters suppress noise at 50 Hz, our goal was to reconstruct the raw EEG recordings as accurately as possible. Simpler methods, such as linear interpolation, might yield similar results after application of the frequency filters."

---

> > ### Comment · Reviewer_dgsY · 2026-06-19
> > **Regarding the changes by the authors**
> >
> > After going through the updated paper, based on the questions I raised. I am satisfied with the authors answers. I am ok with the paper being accepted with the new changes.